



# Sensitivity of aerosol and cloud properties to coupling strength of

# marine boundary layer clouds over the northwest Atlantic

Kira Zeider[1], Kayla McCauley[2+], Sanja Dmitrovic[3], Leong Wai Siu[2], Yonghoon Choi[4,5], Ewan C.
Crosbie[4,5], Joshua P. DiGangi[4], Glenn S. Diskin[4], Simon Kirschler[6,7], John B. Nowak[4], Michael
A. Shook[4], Kenneth L. Thornhill[4,5], Christiane Voigt[6,7], Edward L. Winstead[4,5], Luke D. Ziemba[4],
Paquita Zuidema[8], Armin Sorooshian[1,2,3*]
[1]Department of Chemical and Environmental Engineering, University of Arizona, Tucson, AZ, 85721, USA
[2]Department of Hydrology and Atmospheric Sciences, University of Arizona, Tucson, AZ, 85721, USA
[3]James C. Wyant College of Optical Sciences, University of Arizona, Tucson, AZ 85721, USA
[4]NASA Langley Research Center, Hampton, VA, 23681, USA
[5]Analytical Mechanics Associates, Hampton, VA, 23666, USA
[6]Institute of Atmospheric Physics, German Aerospace Center, Germany
[7]Institute of Atmospheric Physics, University Mainz, Germany
[8]Rosenstiel School of Marine, Atmospheric, and Earth Science, University of Miami, Miami, FL
[+]Now at EPA, Research Triangle Park, NC 27711
*Correspondence to*: Armin Sorooshian (armin@arizona.edu)





**Abstract**
Quantifying the degree of coupling between marine boundary layer clouds and the surface is critical for understanding
the evolution of low clouds and explaining the vertical distribution of aerosols and microphysical cloud properties. In
this study, we use aircraft data from the NASA Aerosol Cloud meTeorology Interactions oVer western ATlantic
Experiment (ACTIVATE) to assess aerosol and cloud characteristics for the following four regimes of coupling
strength as quantified using differences in liquid water potential temperature ($\theta_l$) and total water mixing ratio ($q_t$)
between a near-surface level (~150 m) and directly below cloud bases: strong coupling ($\Delta\theta_l \leq 1.0$ K, $\Delta q_t \leq 0.8$ g kg$^{-1}$
$^{-1}$), moderate coupling with high $\Delta\theta_l$ ($\Delta\theta_l > 1.0$ K, $\Delta q_t \leq 0.8$ g kg$^{-1}$), moderate coupling with high $\Delta q_t$ ($\Delta\theta_l \leq 1.0$ K, $\Delta q_t$
$> 0.8$ g kg$^{-1}$), weak coupling ($\Delta\theta_l > 1.0$ K, $\Delta q_t > 0.8$ g kg$^{-1}$). Results show that (i) turbulence is greater in the strong
coupling regime compared to the weak coupling regime, with the former corresponding to smaller differences in 550
nm aerosol scattering, integrated aerosol volume concentration, and giant aerosol number concentration ($D_p > 3$ µm)
between the near-surface level and just below marine boundary layer (MBL) cloud bases coincident with increased
MBL mixing, (ii) cloud drop number concentration is greater during periods of strong coupling due to the greater
upward vertical velocity and subsequent activation of particles, (iii) sea-salt tracer species (Na$^+$, Cl$^-$, Mg$^{2+}$, K$^+$) are
present in greater concentrations in the strong coupling regime compared to weak coupling, while Ca$^{2+}$, nss-SO$_4^{2-}$,
NO$_3^-$, oxalate, and NH$_4^+$ (tracers of continental pollution) are higher in mass fraction for the weak coupling regime.
Additionally, pH and Cl$^-$:Na$^+$ (a marker for chloride depletion) are consistently lower in the weak coupling regime.
There were differences between the two moderate regimes: the moderate high $\Delta q_t$ regime had greater turbulent mixing
and sea salt concentrations in cloud water, along with smaller differences in integrated volume and giant aerosol
number concentration between the two vertical levels compared. This work shows value in defining multiple coupling
regimes (rather than the traditional coupled versus decoupled) and demonstrates differences in aerosol and cloud
behavior in the MBL for the various regimes.



## 1 Introduction

The composition of marine boundary layer (MBL) cloud properties is strongly linked to the lower troposphere's vertical structure, making it critical to understand the degree of coupling between boundary layer clouds and the ocean's surface. When the MBL is well-mixed, there is a thermodynamic exchange between the ocean's surface and the cloud deck, and it is considered coupled. A decoupled MBL is characterized by a stable layer separating two well-mixed layers (the cloud deck and sub-cloud layer), preventing exchange between the ocean's surface and the cloud base (Nicholls, 1984; Dong et al., 2015; Jones et al., 2011; Wang et al., 2016). Whether the MBL cloud deck is coupled or decoupled to the surface has potentially important implications for cloud and aerosol properties (Dong et al., 2015; Wang et al., 2016; Griesche et al., 2021), radiative forcing (Goren et al., 2018), and precipitation (Bretherton et al., 2010; Dong et al., 2015). Changes in cloud properties and precipitation affect how much solar radiation is reflected to space (Twomey, 1974; Albrecht, 1989), which in turn affects how much radiative cooling occurs (Ramanathan et al., 1989).

Past studies have investigated coupling behavior of marine stratocumulus due to their relatively high frequency over the ocean's surface and strong impact on the Earth's radiation budget (Zuidema et al., 2009; Jones et al., 2011; Dong et al., 2015; Wang et al., 2016; Goren et al., 2018). In marine regions, well-mixed moist thermodynamic statistics indicate coupling of the sub-cloud layer to the surface (Bretherton et al., 1997; Jones et al., 2011; Dong et al., 2015; Wang et al., 2016; Su et al., 2022). Studies beyond those previously mentioned over the southeast and northeast Pacific have applied these methods to other regions, such as the Arctic (Griesche et al., 2021) and over land in the Southern Great Plains of the United States (Su et al., 2022). Table 1 provides a synthesis of previous studies that utilized thermodynamic statistics for determining coupling, including criteria used, the region in which the study was conducted, and the cloud types investigated.

**Table 1: Summary of coupling criteria and regional conditions from previous work in comparison to this study.**

| Study region; reference | Criteria | Secondary criteria | Layers used | Cloud type |
|---|---|---|---|---|
| Southeast Pacific; Jones et al. (2011) | Coupled: $\Delta q_t < 0.5$ g kg$^{-1}$ & $\Delta\theta_\ell < 0.5$ K<br><br>All other profiles are considered decoupled | Coupled: distance between lifting condensation level (LCL) and cloud base is < 150 m<br><br>Decoupled: distance > 150 m | Bottom 25% of surface layer to cloud base height | Marine strato-cumulus |
| Azores (Graciosa Island; Northeast Atlantic); Dong et al. (2015) | | | | |
| Northeast Pacific; Wang et al. (2016) | Decoupled: $\Delta q_t > 0.6$ g kg$^{-1}$ & $\Delta\theta_\ell > 1.0$ K<br><br>All other clouds are considered coupled | N/A | | |



| Southern Great Plains (U.S.); Su et al. (2022) | Coupled: $\Delta\theta_\ell < 1.0$ K  Decoupled: $\Delta\theta_\ell > 1.0$ K | | | Different Thermodynamic Stability (DTDS) method | Cloud base height minus planetary boundary layer height | Low clouds over land, specifically cumulus |
|---|---|---|---|---|---|---|
| | Degree | $\Delta q_t$ | $\Delta\theta_\ell$ | | Below cloud base leg (~100 m below base) minus MinAlt leg (avg. alt ~150 m) | Marine clouds spanning continuum from stratiform to cumulus |
| Northwest Atlantic; This Study | Strong | $\leq 0.8$ g kg$^{-1}$ | $\leq 1.0$ K | N/A | | |
| | Moderate, high $\Delta\theta_\ell$ | $\leq 0.8$ g kg$^{-1}$ | $> 1.0$ K | | | |
| | Moderate, high $\Delta q_t$ | $> 0.8$ g kg$^{-1}$ | $\leq 1.0$ K | | | |
| | Weak | $> 0.8$ g kg$^{-1}$ | $> 1.0$ K | | | |


As over 45% of the ocean's surface is covered by MBL clouds (Warren et al., 1988), examining relations between
aerosol and cloud characteristics with coupling strength is important. Investigation of coupling behavior has not yet
been carried out for the northwest Atlantic region, which is a complex thermodynamic region for such work as it is
not a classical sub-tropical zone with a stratocumulus cloud deck like most regions investigated in Table 1 (Painemal
et al., 2021, 2023). The synoptic conditions over the northwest Atlantic are such that the wintertime has higher cloud
fraction with more influence from stratiform boundary layer clouds, whereas the summertime has more trans-Atlantic
flow in addition to lower cloud fraction with higher sea surface temperatures promoting shallow cumulus clouds
(Painemal et al., 2021). During winter, there is more offshore advection of continental air (Corral et al., 2021;
Dadashazar et al., 2021), enhanced precipitation frequency (Painemal et al., 2021), and cold air outbreaks (CAOs), in
which cold air is advected across the Gulf Stream front resulting in pronounced differences between air and sea surface
temperatures (Brümmer, 1997; Papritz & Spengler, 2015; Seethala et al., 2021). CAOs are typically associated with
strong turbulent mixing, leading to the deepening of the boundary layer (Dadashazar et al., 2021; Painemal et al.,
2021; Papritz & Spengler, 2015; Tornow et al., 2022). During CAO events, surface wind convergence is driven by
horizontal pressure and boundary layer height gradients, contributing to a statically unstable troposphere (Painemal et
al., 2021; Seethala et al., 2021).

Motivated by meteorological differences between the northwest Atlantic and other regions in Table 1, the question
arises as to whether it is restrictive to consider just the categories of coupled and decoupled clouds; instead, it may be
instructive to consider more categories and that they all may have some degree of coupling ranging from weak to
strong. This strategy is built off past reports suggesting that the use of the term "decoupled" may not be appropriate
and that an MBL can be coupled even though it is poorly mixed (Stevens et al., 1998). The latter case can be viewed
as weakly coupled due to episodic updraft-driven convection that is less efficient at mixing the MBL than is the case
in well-mixed MBLs in which downdrafts associated with cloud-top radiative cooling couple the cloud and sub-cloud
layers. Thus, the perspective we embrace in this work is that low-level clouds (< 2 km) can be viewed as always being
coupled to sub-cloud layers but to varying degrees.



The goal of this study is to leverage an opportune aircraft dataset covering multiple seasons between 2020 to 2022
from NASA's Aerosol Cloud meTeorology Interactions oVer the western ATlantic Experiment (ACTIVATE;
Sorooshian et al., 2019) to quantify the frequency of occurrence for four different coupling regimes and how aerosol
and cloud characteristics vary between them. We emphasize that this study is different in nature to those in Table 1 in
that we do not examine as rigorously the vertical extent of the full cloudy boundary layer but instead focus more on
aerosol and cloud characteristics for different coupling regimes based on definitions limited to the vertical region
below cloud bases. The analyses presented here are important for reasons such as knowing how well the aerosol near
the surface level represents the aerosol just below cloud bases, with implications for the aerosol that largely govern
the drop concentration budget. In Sect. 2 we summarize the measurements and methods, including criteria applied
with traditionally used thermodynamic variables to differentiate between four coupling categories. In Sect. 3 we report
results including the frequency of occurrence of the four coupling regimes, and differences in aerosol properties (light
scattering, aerosol number/volume) and cloud microphysical properties (composition and drop number concentration)
between these categories to see how they compare to past studies for other regions. Although there are scarce previous
reports of such findings (e.g., Dong et al., 2015; Wang et al., 2016), the central hypotheses are based on confirming
what has been shown in other regions, in that more strongly coupled cases will have: (i) more similar values for aerosol
properties in the sub-cloud layer compared to closer to the ocean's surface; (ii) cloud composition reflecting
significantly more sea salt influence; and (iii) higher cloud drop number concentration. Differences identified in
aerosol and cloud characteristics between these four coupling regimes are important to inform future research to
account for thermodynamic profiles when examining aspects of aerosol and cloud microphysics when using either
satellite, reanalysis, airborne, or ground-based datasets.
**2 Data and Methodology**
**2.1 Overview of ACTIVATE**
ACTIVATE was largely based out of NASA Langley Research Center in Hampton, Virginia and carried out research
flights (RFs) with two spatially coordinated aircraft as part of six deployments in winter and summer months between
2020 and 2022, with extensive measurement and deployment details provided elsewhere (Sorooshian et al., 2023).
Secondary bases were used for a subset of flights in 2022, including in Bermuda for June 2022. Winter and summer
are broadly defined as including the months of November-April and May-September, respectively. The HU-25 Falcon
flew level legs in, below, and above the MBL to collect in-situ atmospheric state, aerosol, trace gas, and cloud
measurements, while the high-flying King Air at ~9 km launched dropsondes and carried out remote sensing. The
focus of this work is data collected by the Falcon. Out of 179 total flights, 135 were used that offered data conducive
to this study's analysis including having the Falcon conduct "statistical survey" flights with "cloud ensembles" (Fig.
1), along with several physical conditions satisfied as discussed in Sect. 2.5. During statistical survey flights, which
accounted for 93% of ACTIVATE's flights, the Falcon repeatedly flew a series of legs with Fig. 1 visually depicting
one such cloud ensemble whereby the plane flew the following legs in this nominal order: two pairs of legs below
cloud base (BCB) and above cloud base (ACB) followed by a descent to the minimum altitude (MinAlt) possible
(~150 m above sea level) and then a subsequent slant ascent for a leg above cloud top (ACT) followed by a final leg



below cloud top (BCT). Separate ensembles flown in clear air conditions are outside the scope of this work. Each leg
duration was ~3.3 minutes (equivalent to ~ 24 km) with the Falcon flying at ~120 m s$^{-1}$ (Dadashazar et al., 2022a).
The vertical slant ascents/descents between level legs especially down to MinAlt and up to ACT were helpful in
gathering vertically-resolved information during ensembles.

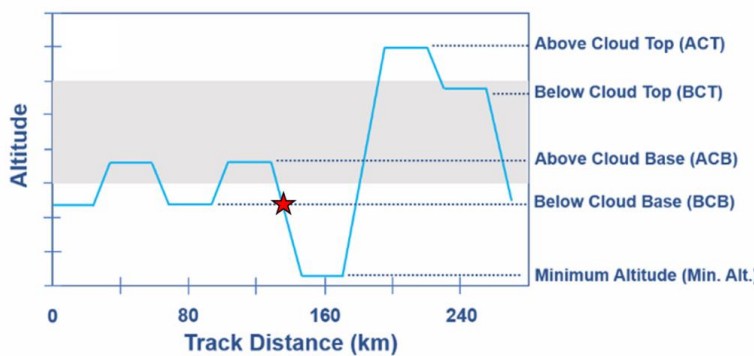


**Figure 1: Cloudy ensemble flight strategy of the HU-25 Falcon during the ACTIVATE flights, where the grey box represents**
**a typical cloud layer with upper and lower boundaries representing cloud top and base, respectively. The order of legs was**
**the nominal plan that was flown, but sometimes the legs were flown in a different order based on flight restrictions and**
**cloud conditions. The red star indicates where the BCB level would be marked for this particular flight pattern, which is**
**during the slant descent from ACB to MinAlt and uses the mean altitude of the preceding BCB leg immediately before the**
**ACB leg; that level would then be compared to the adjacent MinAlt level that begins at the end of the slant descent.**
**Otherwise, MinAlt-BCB pairs that are used include when a MinAlt level leg was immediately preceded or succeeded by a**
**BCB level leg.**

**2.2 Implementation of Flight Legs**
Across ACTIVATE's six deployments, MinAlt and BCB legs were identified for RFs when the Falcon flew cloud
ensembles (Fig. 1). There were several instances when the MinAlt and BCB legs were not immediately adjacent and
separated by another leg, such as at ACB (i.e., the flight order was BCB-ACB-MinAlt; Fig. 1). In those cases, to get
MinAlt/BCB pairs that were closer in time, our method involved identifying the BCB altitude during the slant
altitudinal change (either descent or ascent) between MinAlt-ACB based on the altitude of the BCB leg immediately
before the ACB leg (see red star in Fig. 1). A secondary check was made to ensure that identified BCB leg was below
cloud base using 1-Hz LWC and $N_d$ values from the FCDP (criteria in Sect. 2.5). The vertical structure of the layer
between MinAlt and BCB was examined using data between the last time stamp in the MinAlt/BCB leg (i.e.,
whichever was first in the MinAlt-BCB pair) and first-time stamp in the subsequent BCB/MinAlt leg (i.e., whichever
was second in the MinAlt-BCB pair). For simplicity, we refer to the case of using BCB data during slant profiles as
"legs" too, even though they were not level legs. This study compares various measurement data (Sect. 2.5) between
MinAlt and BCB legs using the last/first 5 seconds of data during adjacent MinAlt-BCB legs, and in the case of slants,
we use the 2 points before and after the actual BCB point (for a total of 5; average altitude range ~ 20 m) to represent
the BCB level with the condition that all data were out of cloud.





**2.3 Instrumentation**
A summary of instrumentation relevant to this study is shown in Table 2 and briefly described here. A nephelometer
(TSI-3563) measured the dry scattering coefficient at 550 nm (particle diameter ($D_p$) < 5.0 μm for 2020 and $D_p$ < 1.0
μm for 2021 and 2022); a Laser Aerosol Spectrometer (LAS; TSI-3340) measured aerosol size distributions (0.1 μm
< $D_p$ < 5.0 μm) and here we use the integrated aerosol volume concentration data; a Fast Cloud Droplet Probe (FCDP;
SPEC Inc.) measured liquid water content (LWC) and particle and cloud drop size distributions between 3 and 50 μm;
a two-dimensional stereo (2D-S; SPEC Inc.; 25 μm < $D_p$ < 1500 μm) probe provided LWC, liquid droplet effective
diameter, and an ice flag, where the ice flag is equal to 1 if ice was detected (otherwise the variable is equal to 0); a
diode laser hygrometer (DLH) measured the water vapor mixing ratio ($q_v$); a turbulent air motion measurement system
(TAMMS) measured three-dimensional winds (Thornhill et al., 2003); and an axial cyclone cloud water collector
(AC3) (Crosbie et al., 2018) collected cloud water samples by inertially separating droplets from the air stream.
Collected cloud water samples were then analyzed post-flight with ion chromatography (IC) with operating conditions
summarized elsewhere (Corral et al., 2022; Gonzalez et al., 2022). Section 2.6 describes the cloud water data in more
detail.

**Table 2: Summary of field campaign instrumentation used and corresponding measurements.**

| Instruments | Measurements | Diameter (μm) | Reference |
|---|---|---|---|
| TSI-3563 Nephelometer | Dry scattering coefficient at 550 nm | < 5.0 for 2020; < 1.0 for 2021 & 2022 | Ziemba et al. (2013) |
| TSI-3340 Laser Aerosol Spectrometer (LAS) | Integrated aerosol volume concentration | 0.1 – 5.0 | Froyd et al. (2019) |
| SPEC Inc. Fast Cloud Drolet Probe (FCDP) | Liquid water content (LWC), particle number concentration ($N_a$), cloud drop number concentration ($N_d$) | 3 – 50 | Kirschler et al. (2022) |
| SPEC Inc. Two-Dimensional Stereo Probe, Horizontal Arm (2DS-H) | LWC, effective diameter for liquid, ice flag | 25 – 1500 | Kirschler et al. (2023) |
| Diode Laser Hygrometer (DLH) | Water vapor mixing ratio ($q_v$) | N/A | Diskin et al. (2002) |
| Axial Cloud Water Collector (AC3) | Cloud water composition | see Sect. 2.6 | Crosbie et al. (2018) |
| Turbulent Air Motion Measurement System (TAMMS) | Three dimensional winds | N/A | Thornhill et al. (2003) |




**2.4 Marine Boundary Layer Coupling**

**2.4.1 Thermodynamic Variables**

To estimate the degree of coupling within the marine boundary layer, we consider the change in vertical profile of two parameters: total water mixing ratio ($q_t$) and liquid water potential temperature ($\theta_\ell$). Relevant to this study are these equations,

$$q_\ell = \frac{LWC}{\rho_d} \tag{1}$$

$$q_t = q_v + q_\ell \tag{2}$$

where the total water mixing ratio is the sum of water vapor mixing ratio ($q_v$) and liquid water mixing ratio ($q_\ell$). The water vapor mixing ratio ($q_v$) provided by the DLH is converted from ppmv to g kg$^{-1}$. The liquid water mixing ratio ($q_\ell$) is defined as the ratio of the mass of liquid water to the mass of dry air within a unit volume of air, which is equivalent to the ratio of LWC (provided by the FCDP) and the density of dry air ($\rho_d$).

Also relevant are these equations,

$$\theta = (T + 273.15) \times \left(\frac{p_0}{p}\right)^\kappa \tag{3}$$

$$\theta_\ell = \theta - \left(\frac{L_v}{c_{pd}}\right) \times q_\ell \tag{4}$$

where in equation 3, T and p are the given temperature in °C and pressure in hPa from Falcon measurements, respectively, $p_0$ is the reference pressure (= 1000 hPa), and kappa is the ratio of gas constant of dry air ($R_d$) to the specific heat of dry air at constant pressure ($c_{pd}$). In equation 4, $L_v$ is latent heat of vaporization and $c_{pd}$ is the specific heat of dry air at constant pressure. When LWC is equal to 0, $\theta_\ell$ is equal to $\theta$. $\theta_\ell$ is useful for the purposes of this study as it is not significantly influenced by evaporating precipitation. Information regarding LWC thresholds for MinAlt-BCB pairs is included in Sect. 2.5.

For each MinAlt and BCB leg, the average $\theta_\ell$ and $q_t$ across the leg was calculated and the difference between the two layers was taken as follows:

$$\Delta q_t = q_{t,MinAlt} - q_{t,BCB^*} \tag{5}$$

$$\Delta \theta_\ell = \theta_{\ell,BCB^*} - \theta_{\ell,MinAlt} \tag{6}$$

where in both equations 5-6, the order of legs on the right-hand side is meant to arrive at a positive value for the difference based on expectation.






### 2.4.2 Coupling Criteria


The criteria we use for the different coupling regimes were informed by (but are not identical to) those used in past
work (Jones et al., 2011; Dong et al., 2015; Wang et al., 2016; Su et al., 2022). Our focus was on comparing the vertical
range between MinAlt and BCB legs due to the focus on examining aerosol characteristics in particular within that
range and also cloud microphysical conditions above cloud base. To qualify as strongly coupled, the difference
between MinAlt and BCB had to satisfy these conditions: $\Delta q_t \leq 0.8$ g kg$^{-1}$ and $\Delta\theta_\ell \leq 1.0$ K (example in Fig. 2a). Since
$\Delta q_t$ is more influenced by evaporation and condensation whereas $\Delta\theta_\ell$ is more affected by air mass mixing (such as
entrainment) and diabatic heating and cooling, it is proposed to have two degrees of moderate coupling – when one
of $\Delta q_t$ and $\Delta\theta_\ell$ fit the strong coupling criteria and the other did not (Fig. 2b-c). Finally, profiles are considered "weakly
coupled" when both $\Delta q_t$ and $\Delta\theta_\ell$ do not satisfy the strong coupling criteria values (Fig. 2d). Vertical profiles of $q_t$ and
$\theta_\ell$ were examined for all MinAlt/BCB pairs to ensure robustness of the categorization method. The use of the two
moderate categories is exploratory in nature and meant to identify if differences are found between both themselves
and the more extreme categories of strong and weak. Appendix A further explores differences between the two
moderate regimes and suggests that the moderately coupled category with high $\Delta\theta_\ell$ is influenced more by processes
above the MBL such as entrainment of dry air with high potential temperature whereas the other moderate category
with high $\Delta q_t$ is influenced by surface processes.

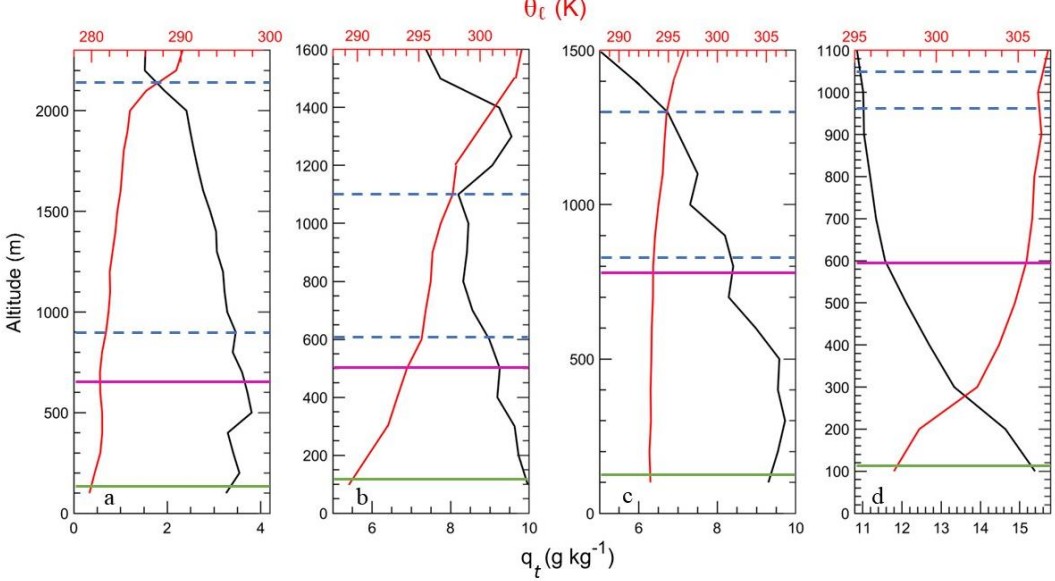


**Figure 2: Representative vertical profiles of $\theta_\ell$ and $q_t$ for (a) strong coupling from RF 44 on 3 February 2021, (b) moderate**

**coupling with high $\Delta\theta_\ell$ from RF 150 on 5 May 2022, (c) moderate coupling with high $\Delta q_t$ from RF 66 on 5 May 2021, and**
**(d) weak coupling from RF 158 on 20 May 2022. The dashed blue lines demarcate the cloud top and base levels, the magenta**



**line indicates the BCB leg, and the green line indicates the MinAlt leg. There was a total of 411 MinAlt-BCB pairs analyzed**
**in this study.**

**2.5 Aerosols and Atmospheric Properties**
There were several aerosol and atmospheric properties investigated in this study: aerosol scattering (scat) at 550 nm
(< 5 μm in 2020 and < 1 μm in 2021-2022), integrated volume (IntV: $0.1 < D_p < 5$ μm), particle number concentration
($N_a$; $3 < D_p < 50$ μm), cloud drop number concentration ($N_d$; $3 < D_p < 50$ μm), and turbulence ($\sigma_w$). Note that the $N_a$
measurement from the FCDP for diameter > 3 μm is important in this study to better isolate sea salt particles (Gonzalez
et al., 2022). The integrated volume also is expected to be influenced by larger sea salt particles in the measurement
size range. These properties were averaged across each MinAlt and BCB pair and the difference between the MinAlt
and BCB values was computed. To account for interference from cloud droplet shatter with the aerosol statistics, we
only looked at MinAlt-BCB pairs when atmospheric conditions for each leg was devoid of cloud, rain, and ice. The
following three criteria had to be met: (1) ice flag from 2DS-H = 0, (2) effective liquid diameter from 2DS-H < 60 μm,
and (3) LWC from FCDP < 0.005 g m$^{-3}$, to filter out conditions with ice, liquid precipitation, and clouds, respectively.
When considering in-cloud conditions for $N_d$, additional criteria were needed based on FCDP data: LWC > 0.05 g m$^{-}$
$^3$ and $N_d > 10$ cm$^{-3}$ (Kirschler et al., 2023). $N_d$ data was collected from ACB legs closest in proximity to a MinAlt-
BCB pair (< 30 min; 60% within 10 min) due to one of the study objectives being to examine how $N_d$ varies between
the four defined coupling regimes. Turbulence was calculated as the standard deviation of the vertical wind velocity
for a level leg as done in other work (e.g., MacDonald et al., 2020).

**2.6 Cloud Water Species**
The nine cloud water species of interest in this study include non-sea salt calcium (nss-Ca$^{2+}$), chloride (Cl$^-$), potassium
(K$^+$), magnesium (Mg$^+$), sodium (Na$^+$), ammonium (NH$_4^+$), nitrate (NO$_3^-$), oxalate, and non-sea salt sulfate (nss-
SO$_4^{2-}$). Calculations of nss-Ca$^{2+}$ and nss-SO$_4^{2-}$ utilized mass ratios and concentrations of pure Ca$^{2+}$, Na$^+$, and SO$_4^{2-}$,
following the methodology outlined in Sect. 2.7 of AzadiAghdam et al. (2019). The IC is used to obtain concentrations
of cloud water species in aqueous units (mg L$^{-1}$), which were then converted to air equivalent concentrations using the
methods described in Gonzalez et al. (2022). Briefly, the cloud water sample was considered in-cloud under the criteria
LWC$_{FCDP}$ > 0.05 g m$^{-3}$. When this condition was met, the concentration was multiplied by the average LWC$_{FCDP}$
measured across the sampling time and divided by the density of water and ultimately converted to μg m$^{-3}$ for the air
equivalent concentration. These units allow one to compare concentrations more fairly between samples to remove
biases due to varying amounts of water in different clouds. As cloud water samples were collected periodically during
flights, samples were only examined when a MinAlt or BCB leg being investigated was within 30 minutes or
overlapped with the collection period. Out of a total of 535 cloud water samples over the 6 deployments, 67 met the
criteria to be used for this study's MinAlt/BCB pairs. Statistics including mean, standard deviation (std. dev.),
minimum, maximum, and quartile ranges were calculated across the 67 data points for all nine cloud water species.

Additionally, cumulative average cloud water mass concentrations and mass fractions were calculated for the 67
samples. The total mass concentration for each coupling regime was found by the summation of only the nine chemical



species investigated in this manuscript. Welch's t-test calculations were conducted to compare the mean concentrations
of the investigated chemical species across coupling regimes. These tests were done in lieu of the traditional t-test due
to the assumption that the data used have unequal variances and thus are slightly more robust.
**3 Results and Discussion**
**3.1 Thermodynamic Criteria**
This section discusses the application of the developed thermodynamic criteria across all MinAlt-BCB pairs. In total,
MinAlt-BCB pairs were investigated (pair locations shown in Fig. 3), with the breakdown of the distribution
across the different degrees of coupling shown in Fig. 4 and Table 3. The majority of the pairs were classified as
strongly coupled, with a breakdown of 71.29% (strongly coupled), 13.63% (moderately coupled with high $\Delta\theta_\ell$),
10.22% (moderately coupled with high $\Delta q_\ell$), and 4.86% (weakly coupled). Strong turbulent mixing in the northwest
Atlantic Ocean, especially during the winter (Brunke et al., 2022) which is when most pairs were identified, is likely
why the majority of pairs were found to be strongly coupled, as the coupling parameters $\theta_\ell$ and $q_\ell$ are relatively constant
vertically from the surface to near cloud bases due to the strong mixing (Fig. 2a).
There are no major spatial distribution differences for MinAlt-BCB pairs across the four coupling regimes with minor
exceptions being that the majority of pairs identified farther offshore around Bermuda were for both the strongly
coupled and moderately coupled with high $\Delta q_\ell$ categories. Also, the strong and moderate coupling with high $\Delta\theta_\ell$
categories had more pairs north of 37.5°N, which coincides with more wintertime sampling of cold air outbreak events
that feature turbulent conditions (e.g., Painemal et al., 2021; Kirschler et al., 2022).

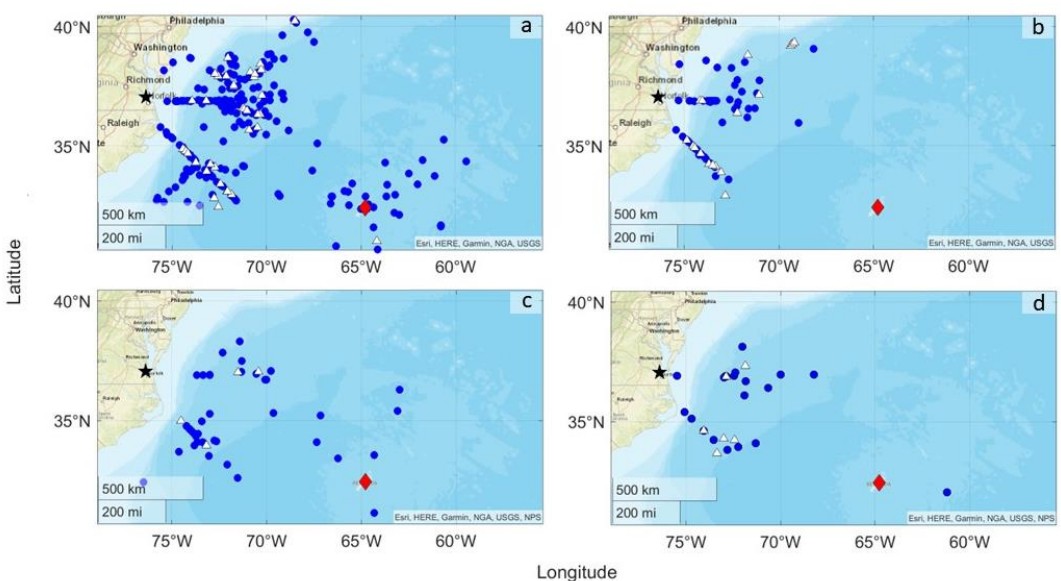




**Figure 3: Locations of the BCB segments of the MinAlt-BCB pairs (blue circles), broken up into the four different degrees of coupling: (a) strong, (b) moderate, high Δθℓ, (c) moderate, high Δqℓ, and (d) weak. The MinAlt legs were close in time to the BCB legs, so only one spatial map is needed to show the approximate data point location for each pair. The locations of the cloud water samples (white triangles) are overlaid on the BCB segment locations. The black star indicates the location of NASA Langley Research Center, and the red diamond indicates Bermuda.**

**Table 3: Total count of MinAlt-BCB pairs categorized into four coupling categories. The winter months include January, February, March, and April, while the summer months include May, June, August, and September.**

|  | Strong coupling | Moderate coupling, high Δθℓ | Moderate coupling, high Δqℓ | Weak coupling |
|---|---|---|---|---|
| All | 293 | 56 | 42 | 20 |
| Summer | 118 | 12 | 18 | 11 |
| Winter | 175 | 44 | 24 | 9 |

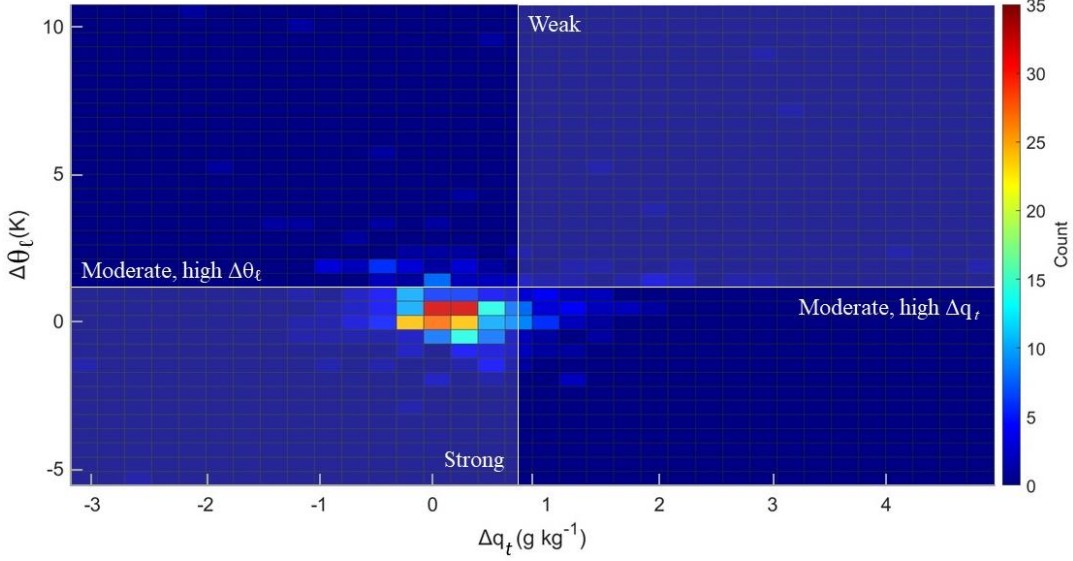

**Figure 4: Joint frequency histogram of Δθℓ versus Δqℓ for MinAlt-BCB pairs, categorized into four coupling regimes. There is a total of 411 pairs plotted (strong coupling = 293, moderate coupling with high Δθℓ = 56, moderate coupling with high Δqℓ = 42, weak coupling = 20). Figure S1 is a similar scatterplot that breaks down the MinAlt-BCB pairs into seasons.**

Figure S1 shows the seasonal distribution of the MinAlt-BCB pairs from summer and winter deployments (counts provided in Table 3). There were more MinAlt-BCB pairs during the winter versus summer (252 vs. 159) largely due to the greater ease of sampling such cases with the higher wintertime cloud fraction in the region (Painemal et al., 2021; Kirschler et al., 2022, 2023). But generally, the distribution of coupling categories was the same (summer/winter): 74.21/69.44% strongly coupled, 7.55/17.47% moderately coupled with high Δθℓ, 11.32/9.52%



moderately coupled with high $\Delta q_t$, and 6.92/3.57% weakly coupled. The frequency of the moderately coupled with
high $\Delta q_t$ category was relatively higher in summer versus winter compared to the other moderate category, which is
coincident with the summer having higher temperatures (i.e., higher $q_v$) and more flights farther south in Fig. 3 where
temperatures are warmer as compared to farther north.

For context, applying the criteria from past work in Table 1 (Jones et al., 2011; Dong et al., 2015) to this dataset (i.e.,
$\Delta q_t$ [g kg$^{-1}$] and $\Delta \theta_\ell$ [K] < 0.5 for coupled and all others decoupled) would have led to 206 and 205 coupled and
decoupled cases, respectively, with a seasonal breakdown as follows (summer/winter): 42.72/57.28% coupled,
34.63/65.37% decoupled. However, we caution that the compared vertical levels differ between these studies. For
example, Jones et al. (2011) compared levels encompassing more of the full extent of the cloudy MBL (e.g., somewhat
analogous to the use of MinAlt and BCT in Fig. 1), whereas in this study we compare MinAlt to BCB due to our focus
on aerosol characteristics, which are difficult to measure in clouds. Also, the frequency of occurrence of the four
coupling regimes in this study are driven in part by how flights were designed to fly towards areas with relatively
higher cloud fraction without complicating scenes such as with multiple cloud layers; thus, the results in Table 3 and
Figure 4 for instance are not a fully accurate depiction of the actual frequency of occurrence over the northwest Atlantic
but rather more of a summary of what was experienced during ACTIVATE flights.

**3.2 Aerosol and Atmospheric Properties**
The results of the aerosol and atmospheric parameter calculations across the four different coupling regimes are
provided in Table 4 (seasonal results in Tables S1-S2). Of the 411 MinAlt-BCB pairs, 293 were used in aerosol
calculations after eliminating pairs that may have been influenced by rain, cloud, or ice interference. As a note, when
quantifying altitudinal differences in variables across different coupling regimes, the mean at each altitude is used as
the comparison parameter unless otherwise stated, as outliers were already removed prior to data analysis.

The first hypothesis of this study is that strongly coupled regimes would have greater turbulence ($\sigma_w$) than weakly
coupled regimes. This hypothesis is confirmed when examining $\sigma_w$ results at both MinAlt (strong/weak = 0.86/0.55
m s$^{-1}$) and BCB levels (strong/weak = 0.70/0.49 m s$^{-1}$). The two categories of moderate coupling had greater turbulence
at both altitudes compared to weak coupling, and sometimes had greater turbulence than pairs categorized as strongly
coupled. This suggests considering multiple coupling regimes for the northwest Atlantic is important to tease out such
nuances as differences in the thermodynamic profiles can potentially coincide with different aerosol and cloud
characteristics as discussed subsequently.

**Table 4: Statistics for various atmospheric properties investigated across the MinAlt-BCB pairs (Δ calculation refers to the**
**MinAlt value minus the BCB value), except for MinAlt $\sigma_w$ and BCB $\sigma_w$, which are the average $\sigma_w$ for each respective leg**
**and for $N_d$, which is calculated in ACB legs. Each property is broken down into the different degrees of coupling (n = number**
**of points used in each coupling category). Variable acronyms defined in Sect. 2.5.**

| Degree of Coupling | Mean | Std. Dev. | Min | 25% | 50% | 75% | Max | n |
|---|---|---|---|---|---|---|---|---|



| | | | | | | | | | |
|---|---|---|---|---|---|---|---|---|---|
| $\Delta$scat | Strong | 2.2 | 2.1 | 0.00 | 0.78 | 1.7 | 2.8 | 13.9 | 274 |
| | Moderate, high $\Delta\theta_\ell$ | 3.5 | 3.5 | 0.07 | 0.97 | 2.4 | 4.6 | 14.6 | 52 |
| | Moderate, high $\Delta q_t$ | 2.4 | 2.1 | 0.01 | 0.70 | 1.8 | 3.6 | 9.2 | 39 |
| | Weak | 3.5 | 3.3 | 0.01 | 0.88 | 2.2 | 6.6 | 10.8 | 20 |
| $\Delta$IntV | Strong | 2.5 | 2.6 | 0.02 | 0.67 | 1.7 | 3.5 | 13.6 | 288 |
| | Moderate, high $\Delta\theta_\ell$ | 2.1 | 2.2 | 0.00 | 0.46 | 1.5 | 2.9 | 9.3 | 54 |
| | Moderate, high $\Delta q_t$ | 1.9 | 1.9 | 0.01 | 0.46 | 1.2 | 2.8 | 8.3 | 41 |
| | Weak | 2.8 | 2.3 | 0.18 | 1.0 | 2.4 | 4.2 | 7.6 | 20 |
| $\Delta N_{a>3\mu m}$ | Strong | 0.32 | 0.55 | 0.00 | 0.05 | 0.13 | 0.35 | 4.9 | 288 |
| | Moderate, high $\Delta\theta_\ell$ | 0.33 | 0.64 | 0.00 | 0.03 | 0.13 | 0.31 | 3.6 | 54 |
| | Moderate, high $\Delta q_t$ | 0.15 | 0.14 | 0.00 | 0.06 | 0.11 | 0.21 | 0.61 | 41 |
| | Weak | 0.53 | 1.5 | 0.01 | 0.02 | 0.08 | 0.27 | 5.9 | 20 |
| $N_d$ | Strong | 344 | 217 | 19 | 193 | 310 | 473 | 954 | 238 |
| | Moderate, high $\Delta\theta_\ell$ | 419 | 242 | 45 | 228 | 374 | 610 | 962 | 48 |
| | Moderate, high $\Delta q_t$ | 329 | 154 | 25 | 235 | 327 | 430 | 671 | 31 |
| | Weak | 275 | 181 | 50 | 107 | 245 | 411 | 606 | 18 |
| MinAlt $\sigma_w$ | Strong | 0.86 | 0.49 | 0.00 | 0.47 | 0.79 | 1.2 | 2.4 | 293 |
| | Moderate, high $\Delta\theta_\ell$ | 1.0 | 0.57 | 0.00 | 0.52 | 1.1 | 1.3 | 2.2 | 56 |
| | Moderate, high $\Delta q_t$ | 0.81 | 0.44 | 0.00 | 0.51 | 0.73 | 0.99 | 1.9 | 42 |
| | Weak | 0.55 | 0.38 | 0.00 | 0.19 | 0.51 | 0.90 | 1.3 | 20 |
| BCB $\sigma_w$ | Strong | 0.70 | 0.62 | 0.00 | 0.26 | 0.60 | 1.0 | 4.0 | 293 |
| | Moderate, high $\Delta\theta_\ell$ | 0.64 | 0.62 | 0.00 | 0.00 | 0.53 | 1.1 | 2.2 | 56 |
| | Moderate, high $\Delta q_t$ | 0.81 | 0.76 | 0.00 | 0.26 | 0.71 | 1.1 | 3.3 | 42 |
| | Weak | 0.49 | 0.50 | 0.00 | 0.04 | 0.30 | 0.87 | 1.6 | 20 |
| BCB - MinAlt $\sigma_w$ | Strong | -0.15 | 0.66 | -2.0 | -0.45 | -0.16 | 0.10 | 3.6 | 285 |
| | Moderate, high $\Delta\theta_\ell$ | -0.34 | 0.57 | -2.2 | -0.81 | -0.23 | 0.10 | 0.59 | 53 |
| | Moderate, high $\Delta q_t$ | 0.01 | 0.74 | -1.6 | -0.28 | -0.08 | 0.20 | 2.9 | 42 |
| | Weak | -0.05 | 0.47 | -1.1 | -0.35 | 0.01 | 0.25 | 0.90 | 20 |


The second hypothesis is that aerosol scattering ($\Delta$scat), integrated volume ($0.1 < D_p < 5$ µm; $\Delta$IntV), and giant particle number concentration ($3 < D_p < 50$ µm; $\Delta N_{>3\mu m}$) would have more homogenous concentrations (i.e., smaller MinAlt-BCB differences) in strongly coupled regimes compared to weakly coupled regimes due to greater mixing for the former as supported by the higher $\sigma_w$ results already shown. This hypothesis is supported (Table 4) since strong coupling cases exhibited lower mean differences (MinAlt-BCB) than weak coupling ($\Delta$scat: 2.2/3.5 Mm[-1], $\Delta$IntV: 2.5/2.8 µm[3] cm[-3], and $\Delta N_{>3\mu m}$: 0.3/0.5 cm[-3], for strong/weak regimes). The third hypothesis was that cloud drop number concentration ($3 < D_p < 50$ µm; $N_d$) would be greater in strong coupling conditions, as stronger updrafts and turbulence would help to activate more particles into cloud droplets (this was also found in Dong et al., 2015). This is confirmed in Table 4: mean $N_d$: 344/275 cm[-3] for strong/weak regimes for ACB legs coinciding with each MinAlt-BCB pair. This result is consistent with past studies for the northwest Atlantic linking stronger turbulence to greater droplet activation efficiency (Kirschler et al., 2022; Dadashazar et al., 2021). The results based on medians agree with those of mean values in Table 4.

When comparing the moderate coupling regimes with the strong and weak regimes, neither $\Delta$scat, $\Delta$IntV, nor $\Delta N_{>3\mu m}$ showed a consistent trend in terms of being higher or lower across all three variables. However, one consistent feature among the moderate regimes is that the moderate high $\Delta q_t$ category showed smaller differences than moderate high



$\Delta\theta_\ell$ for the three aerosol variables. Sometimes, the difference calculations with the lowest values did not occur during
the strong coupling cases, but rather during moderate coupling with high $\Delta q_t$ cases (i.e., $\Delta IntV = 1.9$ $\mu m^3$ $cm^{-3}$, $\Delta N_{>3\mu m}$
$= 0.2$ $cm^{-3}$). These low differences presumably should coincide with the highest values of $\sigma_w$. This is somewhat
supported by how BCB $\sigma_w$ was greatest for the moderate coupling with high $\Delta q_t$ regime ($0.81 \pm 0.76$ m $s^{-1}$), although
MinAlt $\sigma_w$ was greatest for the moderate coupling with high $\Delta\theta_\ell$ regime ($1.00 \pm 0.57$ m $s^{-1}$) with the value for the
moderate coupling with high $\Delta q_t$ regime being $0.81 \pm 0.44$ m $s^{-1}$. Also, Appendix A provides discussion in support of
why the high $\Delta q_t$ category may have small aerosol differences between MinAlt and BCB levels, whereby surface
effects may be at play to help promote mixing in the MBL. Interestingly, the highest $N_d$ values were for the moderate
high $\Delta\theta_\ell$ category with a mean of 419 $cm^{-3}$, which can partly be explained by how most of these cases occurred during
the winter flights when $N_d$ is higher than in the summer (see also Tables S1 and S2) due to strong updraft velocities
that efficiently activate particles into droplets (e.g., Kirschler et al., 2022). These conditions in winter were common
during cold air outbreaks (Dadashazar et al., 2021).

**3.3 Cloud Water Species**
67 cloud water samples were used in this study (Table 5), with 59.70% of the samples falling into the strong coupling
regime, followed by moderate coupling with high $\Delta\theta_\ell$ (25.37%), weak coupling (8.96%), and lastly moderate coupling
with high $\Delta q_t$ (5.97%). Locations of samples are shown in Fig. 3. Within the strong coupling and moderate coupling
with high $\Delta\theta_\ell$ categories, there were several samples north of 37°N, whereas the moderate coupling with high $\Delta q_t$ and
weak coupling samples were all south of that latitude. The former two categories include substantially more data
during the winter when air masses typically come from the continent featuring urban emissions (Dadashazar et al.,
2022a).

**Table 5: Average cloud water mass concentrations (μg m$^{-3}$) and mass fractions (in %) for all cloud water samples. Also**
**shown is the pH and Cl$^-$:Na$^+$ mass ratio. Results are categorized into different degrees of coupling, and the ratio of weak-**
**to-strong coupling is also reported.**

| | Strong | Moderate, high $\Delta\theta_\ell$ | Moderate, high $\Delta q_t$ | Weak | Weak : Strong |
|---|---|---|---|---|---|
| Mass concentration (μg m$^{-3}$) | | | | | |
| **Total** | **87.45** | **57.93** | **91.92** | **10.32** | **0.12** |
| Cl$^-$ | 45.6 | 32.0 | 52.2 | 4.1 | 0.09 |
| Na$^+$ | 27.9 | 18.6 | 29.9 | 2.5 | 0.09 |
| Mg$^{2+}$ | 3.33 | 2.22 | 3.59 | 0.35 | 0.10 |
| K$^+$ | 0.56 | 0.37 | 0.59 | 0.05 | 0.09 |
| nss-Ca$^{2+}$ | 0.53 | 0.18 | 0.17 | 0.03 | 0.06 |
| nss-SO$_4^{2-}$ | 2.6 | 1.5 | 1.7 | 1.2 | 0.47 |
| NO$_3^-$ | 6.0 | 2.7 | 3.3 | 1.5 | 0.26 |





| | | | | | |
|---|---|---|---|---|---|
| Oxalate | 0.10 | 0.01 | 0.03 | 0.01 | 0.14 |
| $NH_4^+$ | 0.89 | 0.37 | 0.33 | 0.56 | 0.63 |
| Mass fraction (%) | | | | | |
| $Cl^-$ | 52.11 | 55.21 | 56.79 | 39.48 | 0.76 |
| $Na^+$ | 31.94 | 32.07 | 32.56 | 24.31 | 0.76 |
| $Mg^{2+}$ | 3.81 | 3.84 | 3.91 | 3.38 | 0.89 |
| $K^+$ | 0.64 | 0.63 | 0.65 | 0.48 | 0.75 |
| nss-$Ca^{2+}$ | 0.60 | 0.32 | 0.19 | 0.31 | 0.51 |
| nss-$SO_4^{2-}$ | 2.93 | 2.65 | 1.90 | 11.56 | 3.95 |
| $NO_3^-$ | 6.86 | 4.63 | 3.63 | 14.95 | 2.18 |
| Oxalate | 0.11 | 0.01 | 0.04 | 0.13 | 1.20 |
| $NH_4^+$ | 1.02 | 0.63 | 0.35 | 5.41 | 5.32 |
| $Cl^-$:$Na^+$ mass ratio and pH | | | | | |
| pH | 4.92 | 4.60 | 5.29 | 4.44 | 0.90 |
| $Cl^-$:$Na^+$ | 1.65 | 1.66 | 1.69 | 1.53 | 0.92 |
| n | 40 | 17 | 4 | 6 | |

Figure 5 provides composition statistics for the cloud water samples categorized into the four coupling regimes. As a note, the notches (and shading) of the box plots help to compare air-equivalent mass concentration medians across the different coupling categories and aid in the determination of statistical significance. Since this study has utilized means instead of medians when comparing values across coupling regimes, mean concentrations are provided in the SI (Tables S3-S5) and the results of Welch's t-tests for each category are given in Table S6. This study also investigated cumulative average mass concentrations and mass fractions (Table 5) to paint a clearer picture of the breakdown of chemical species for different degrees of coupling. Based on previous work for stratocumulus clouds over the northeast Pacific (Wang et al., 2016), we hypothesize that samples from strong coupling regimes would have higher mass concentrations compared to weakly coupled regimes owing to higher concentrations of sea salt constituents (e.g., $Cl^-$, $Na^+$, $Mg^+$, and $K^+$). More turbulent conditions in strongly coupled cases are thought to promote more mixing of sea salt into boundary layer clouds, which can be detected with cloud water composition measurements (e.g., Dadashazar et al., 2017).





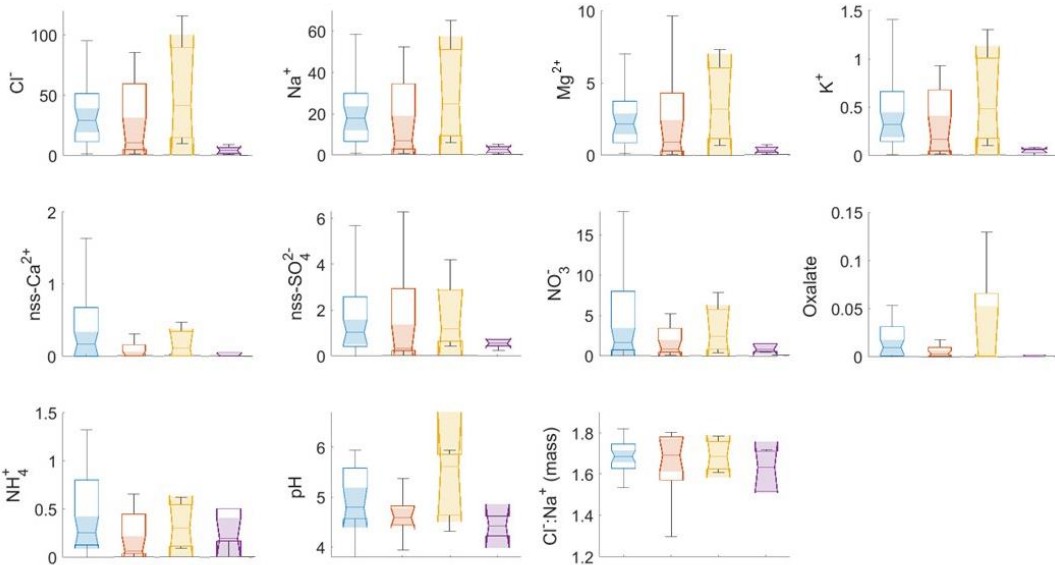

**Figure 5: Notched box plots of species concentrations (µg m⁻³), Cl⁻:Na⁺ mass ratio, and pH from cloud water samples collected during periods coinciding with MinAlt-BCB pairs. The box plots are colored according to degree of coupling: blue (strong), red (moderate, high $\Delta\theta_l$), yellow (moderate, high $\Delta q_t$), and purple (weak). The notches of the boxes assist in the determination of statistical significance between multiple medians (the shading indicates where the notches begin and end). If notches/shading do not overlap, the medians are statistically different from one another (also referred to as statistically significant). Table S6 provides the results of Welch's t-tests, which compares the means of two groups and determines if they are statistically different. The tests were performed on the mean cloud water concentrations of these nine chemical species, pH, and Cl⁻:Na⁺.**

Strong coupling regime samples exhibit higher average mass concentrations compared to weak coupling, and $Na^+$, $Cl^-$, $K^+$, and $NO_3^-$ were all found to be statistically different across the two regimes (p-values: $1.07E^{-4}$, $3.54E^{-5}$, $5.52E^{-5}$, and $9.57E^{-3}$, respectively). The most abundant species by mass across the coupling regimes were usually $Cl^-$, $Na^+$, and $NO_3^-$, similar to the results of Wang et al. (2016). Further, although lower in absolute mass concentration, some species were relatively more abundant (i.e., higher mass fraction) in the weak coupling regime: $nss\text{-}SO_4^{2-}$ (11.56% [weak] vs. 2.93% [strong]), $NO_3^-$ (14.95% [weak] vs. 6.86% [strong]), $NH_4^+$ (5.41% [weak] vs. 1.02% [strong]). Oxalate was very low in overall concentration and exhibited comparable mass fractions: 0.13% [weak] vs. 0.11% [strong]. These results are consistent with the idea of surface emissions (mainly sea salt) driving cloud water composition in turbulent conditions (i.e., strongly coupled) in contrast to weakly coupled clouds that have much lower overall mass concentrations of the reported ions but relatively more influence from non-sea salt species. The two moderate coupling regimes include samples with concentration and mass fraction values more similar to the strongly coupled regime, with even higher sea salt tracer species concentrations for the moderate high $\Delta q_t$ regime. This is consistent with the aerosol results in Sect. 3.2 that suggested this latter category can have appreciable influence from the surface.



Wang et al. (2016) analyzed 35 cloud water constituents for northeast Pacific stratocumulus clouds and found that 27
chemical species were higher in coupled clouds, with the remaining eight (acetate, formate, Si, $NO_2^-$, Al, Mn, Cr, and
Co) higher in decoupled clouds due to relatively more continental influence. The 27 cloud water species that were
higher in coupled clouds were associated with a mix of anthropogenic and natural sources (i.e., sea salt emissions for
$Cl^-$ and $Na^+$). Conversely, the eight species that were higher in decoupled clouds were associated with crustal matter
and biogenic sources. Several of the most abundant species from our study ($Na^+$, $Cl^-$, $Mg^{2+}$) are common sea salt
tracers, while $NO_3^-$ sources in the region may include ocean sea spray and biogenic emissions, wildfires, agricultural
emissions, and ship exhaust (Corral et al., 2020; Corral et al., 2021; Corral et al., 2022; Shah et al., 2018). Note that
nitric acid can partition effectively into cloud droplets as well, which can drive up cloud water $NO_3^-$ levels (e.g.,
Prabhakar et al., 2014). At least some of the species with higher mass fractions in the weakly coupled regime (e.g.,
nss-$SO_4^{2-}$, $NO_3^-$) have previously been linked to combustion sources in this region, such as industrial emissions and
transportation (Brock et al., 2008; Song et al., 2001). Ammonium is a major base forming salts with nss-$SO_4^{2-}$ and
$NO_3^-$, whereas oxalate has diverse sources (e.g., continental, marine) and can be associated with sea salt and produced
via cloud processing (e.g., Stahl et al., 2020; Hilario et al., 2021). Nss-$Ca^{2+}$ is often associated with continental crustal
matter (Ma et al., 2021; Edwards et al., 2024), and its concentrations are generally very low. Similar to oxalate, these
suggest a low influence of dust during the majority of ACTIVATE flights.

In addition to examining mass concentrations, we also examined pH and the $Cl^-$:$Na^+$. Regarding the latter ratio, sea
salt chloride concentrations can be reduced in the presence of acidic species such as sulfuric and nitric acids (e.g.,
Braun et al., 2017; Edwards et al., 2024). This phenomenon is known as $Cl^-$ depletion, and it can be calculated by
taking the ratio of $Cl^-$:$Na^+$. For context, Wang et al. (2016) reported no major difference in the $Cl^-$:$Na^+$ ratio in cloud
water over the northeast Pacific but measured lower pH in coupled clouds (4.26) versus decoupled clouds (4.48). In
this study, samples in the weak coupling regime exhibited the lowest pH (4.4 vs. 5.0 for strong coupling) and $Cl^-$:$Na^+$
(1.5 vs. 1.7 for strong coupling), which potentially could be related to the higher relative amount of nss-$SO_4^{2-}$, $NO_3^-$.
The two moderate coupling regimes feature samples with pH and $Cl^-$:$Na^+$ values more similar to strongly coupled
samples.

A limitation in this analysis is that there were only six cloud water samples that fell into the weak coupling regime.
Future work examining the sensitivity of aerosol and cloud characteristics to coupling regimes should try to obtain
better sampling coverage across all regimes.

**4 Conclusions**
This study used data collected during the NASA ACTIVATE mission (2020–2022) from the HU-25 Falcon to assess
the frequency of different degrees of MBL cloud coupling and also how aerosol and cloud characteristics varied among
four such regimes. MinAlt and BCB legs were used to assess thermodynamic statistics along with turbulence, aerosol,
and cloud variables, which were calculated at each leg and the differences of the two legs were taken for final





comparison metrics. Cloud water species and $N_d$ values associated with MinAlt and BCB pairs were analyzed when
cloud sampling occurred within 30 minutes of a MinAlt-BCB pair.

Vertical profiles between MinAlt and BCB pairs were divided into four degrees of coupling: strongly coupled ($\Delta q_t \leq$
0.8 g kg$^{-1}$, $\Delta\theta_\ell \leq 1.0$ K), moderately coupled with high $\Delta\theta_\ell$ ($\Delta q_t \leq 0.8$ g kg$^{-1}$, $\Delta\theta_\ell > 1.0$ K), moderately coupled with
high $\Delta q_t$ ($\Delta q_t > 0.8$ g kg$^{-1}$, $\Delta\theta_\ell \leq 1.0$ K), and weakly coupled ($\Delta q_t > 0.8$ g kg$^{-1}$, $\Delta\theta_\ell > 1.0$ K). In total, 411 MinAlt-BCB
pairs were investigated, along with 67 cloud water samples. Using this coupling categorization criteria, only a handful
of weakly coupled MBL clouds were detected (20, compared to 286 with strong coupling). The relative amounts of
the regimes did not vary substantially between the winter and summer seasons. Instead, particular focus was placed
on comparing regimes with strong coupling to those with weak coupling. Support for the coupling criteria was sought
through five different aerosol/cloud/dynamic parameters ($\Delta$scat, $\Delta$IntV, $\Delta N_{>3\mu m}$, $N_d$, and $\sigma_w$) and 11 cloud water
variables (nss-Ca$^{2+}$, Cl$^-$, K$^+$, Mg$^+$, Na$^+$, NH$_4^+$, NO$_3^-$, oxalate, nss-SO$_4^{2-}$, pH, Cl$^-$:Na$^+$). Turbulence was generally greater
during regimes of strong coupling compared to weak coupling, which corresponded to lower values of $\Delta$scat, $\Delta$IntV,
and $\Delta N_{>3\mu m}$ due to better presumed mixing in the MBL. $N_d$ was higher for strong coupling regimes, as higher
turbulence likely encouraged more cloud drop activation, which was also observed in Dong et al. (2015) for the
northeast Atlantic. Sea salt tracers (e.g., Na$^+$, Cl$^-$, and K$^+$) were higher in concentration in strongly coupled compared
to weakly coupled MBL clouds and were found to have statistically significant differences across the two coupling
regimes. Additionally, nss-SO$_4^{2-}$, NO$_3^-$, and NH$_4^+$ which are linked to continental sources, were found in higher mass
fractions during weak coupling regimes, which was also observed in Wang et al. (2016) for northeast Pacific
stratocumulus clouds, corresponding to lower values of both cloud water pH and the Cl$^-$:Na$^+$ ratio.

The inclusion of two moderate coupling categories is shown to be insightful as differences between the two potentially
can be explained by the relative influence of subsidence/entrainment versus surface effects. More specifically, the
moderately coupled category with high $\Delta\theta_\ell$ is thought to be influenced more by processes above the MBL such as
entrainment of dry air with high potential temperature whereas the other moderate category with high $\Delta q_t$ likely has
more influence from surface processes. These speculations are supported by how the moderate high $\Delta q_t$ regime
exhibited even more turbulent mixing than the strong coupling regime, yielding the highest sea salt concentrations in
cloud water and the lowest values of $\Delta$IntV and $\Delta N_{>3\mu m}$. Furthermore, the moderate high $\Delta\theta_\ell$ category exhibited the
highest mean $N_d$ value (419 cm$^{-3}$) of any category (275-344 cm$^{-3}$ for the other three categories), which can be explained
partly by how most of these cases (44 of 56) were in winter flights when $N_d$ is typically higher than summer, especially
during cold air outbreaks (e.g., Dadashazar et al., 2021).

This study is the first to our knowledge to investigate degrees of coupling in MBL clouds through thermodynamic
statistics in the northwest Atlantic with a focus on aerosol and cloud microphysical characteristics. Further research
of this nature is needed in other regions to assess thermodynamic criteria for MBL cloud to surface coupling, including
how aerosol and cloud characteristics change with degrees of MBL coupling in different regions. The results here
indicate that a failure to account for different coupling regimes can mix together varying aerosol and cloud





microphysical characteristics in data analysis studies, which increases risk of separating out important details such as
how cloud composition is very different across the spectrum of cloud coupling strength. A limitation of this study to
build on is obtaining more statistics for the more weakly coupled category, which in part may be influenced by how
flight plans are designed. The results of this research have important implications for studies of aerosol-cloud
interactions, as not considering coupling strength will make interpretations difficult, as we have shown important
differences for aerosol and cloud properties.

**Appendix A. Discussion of the two moderate regimes**
To help with the interpretation of the two moderate regimes defined in Table 1, we provide a perspective based on the
following discussion. Using equation 1 but expanding it to take the difference of the liquid water potential temperature
between the BCB and MinAlt flight legs yields the following:
$\Delta\theta_\ell = \left(\theta_{BCB} - \left(\frac{L_v}{c_{pd}}\right) \times q_{\ell,BCB}\right) - \left(\theta_{MinAlt} - \left(\frac{L_v}{c_{pd}}\right) \times q_{\ell,MinAlt}\right)$   (A1)
Note that both $q_\ell$ terms are small below cloud. The $\Delta\theta_\ell$ value can thus be large due to large-scale subsidence or
entrainment when dry air from free troposphere with high $\theta$ which is potentially mixed with air at the BCB level.
While $\theta_{MinAlt}$ can be high due to surface heating, it acts to reduce $\Delta\theta_\ell$. Also, the current MinAlt is slightly above the
typical surface layer and hence the surface inversion. Also, note that:
$\Delta q_t = (q_{v,MinAlt} + q_{l,MinALt}) - (q_{v,BCB} + q_{l,BCB})$   (A2)
where both $q_l$ terms are small below cloud and are typically much smaller than $q_v$. The range of $q_v$ is largely controlled
by the temperature due to the Clapeyron-Clausius equation (the higher temperature, the higher saturation vapor
pressure). While high $\Delta q_t$ may be due to low $q_v$ at the BCB level, it is more likely due to high $q_v$ near the surface
because saturation vapor pressure exponentially increases with temperature.
To conclude, the $\Delta\theta_\ell$ term is more likely influenced by features above the MBL while the $\Delta q_t$ term is more likely
influenced by near-surface effects.

**Data availability**
The ACTIVATE dataset can be downloaded at https://doi.org/10.5067/SUBORBITAL/ACTIVATE/DATA001
(ACTIVATE Science Team, 2020).

**Author contributions**
YC, EC, JPD, GSD, MAF, SK, JBN, MAS, KLT, CV, ELW, and LDZ collected and/or prepared the data. KTZ, SD,
and KM conducted data analysis. KTZ, KM, and SD conducted the formal investigation. KTZ, LWS, and AS
conducted data interpretation. KTZ and AS prepared the manuscript with editing from all co-authors.

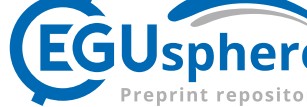

**Competing interests**

At least one of the (co-)authors is a member of the editorial board of Atmospheric Chemistry and Physics.

**Disclaimer**

Publisher's note: Copernicus Publications remains neutral with regard to jurisdictional claims in published maps and institutional affiliations.

**Acknowledgements**

We thank the pilots and aircraft maintenance personnel of NASA Langley Research Services Directorate for conducting ACTIVATE flights and all others who were involved in executing the ACTIVATE campaign.

**Financial support**

ACTIVATE is a NASA Earth Venture Suborbital-3 (EVS-3) investigation funded by NASA's Earth Science Division and managed through the Earth System Science Pathfinder Program Office. University of Arizona investigators were supported by NASA grant no. 80NSSC19K0442 and ONR grant no. N00014-21-1-2115. CV and SK were funded by DFG SPP-1294 HALO under project no. 522359172 and by the European Union's Horizon Europe program through the Single European Sky ATM Research 3 Joint Undertaking projects CONCERTO (grant no 101114785) and CICONIA (grant no 101114613).





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
