# Peer review of "Sensitivity of aerosol and cloud properties to coupling strength of"

_EGUsphere, 2024_

## Referee Comment (RC1)

Review of *Sensitivity of aerosol and cloud properties to coupling strength of marine boundary layer clouds over the northwest Atlantic* by Zeider et al. (2024)

**Suggestion: Major Revision**

Zeider et al. use aircraft measurements from NASA's ACTIVATE field campaign between 2020 and 2022 to investigate aerosol and cloud properties in the marine boundary layer (MBL) for four different coupling regimes. The authors analyze what they refer to as strongly, weakly and two sets of moderately coupled MBLs, instead of only differentiating coupled and decoupled MBLs. These coupling regimes are based on potential temperature and total water mixing ratio differences between cloud base and minimum flight altitude. The authors present findings that are novel for the investigated region and their results highlight the importance of investigating the degree of coupling.

Overall, the paper is well-written, easy to follow, concise and has important findings regarding coupling in the marine boundary layer. Thus, I believe that the manuscript merits publication. Nevertheless, there are some comments mostly regarding the robustness of the results that should be addressed by the authors before publication.

**Major comments**

1. **Impact of measurement uncertainty on results:** Figure S1 shows that many data points are very close to the boundary of their regime. Given measurement uncertainties it is possible that some data points belong to a different regime. I suggest that the authors test the sensitivity of their results by varying the $\Delta\theta_l$ and $\Delta q_t$ values (leading to different regimes for some data) based on measurement uncertainty. Other approaches to account for the measurement uncertainty would be possible as well. I want to add that I found Figure S1 more informative than Figure 4. If the authors want to retain Figure 4, I suggest using white color for pixels with no data points.

2. **Missing significance testing in section 3.2:** In section 3.3 the authors test the statistical significance of their results, however, this is missing in section 3.2. For some parameters, standard deviations are relatively large and in general the sample sizes of the non-strongly coupled regimes is quite small. I suggest conducting statistical significance tests for the values in Table 4. At least for the comparisons done in the text some level of statistical significance should be mentioned. Instead of displaying the data in Table 4, the authors could also consider presenting this data similar to the display shown in Figure 5.

3. **Moderate coupling cases with very large $\Delta\theta_l$:** Figure S1 and 4 show moderately coupled cases with $\Delta\theta_l$ of 5-10 K. For such a substantial difference of $\theta_l$ can one still

speak of a (moderately) coupled boundary layer? I guess what I am wondering is whether the data from these (admittedly) few points is consistent with the rest of the data in this coupling regime?

**Specific Comments**

4. While most of the paper is well-written, I found the abstract a bit hard to read on the first read. There a very long and wordy sentences. It might make sense to revisit the abstract and edit it for easier readability.
5. $\Delta\theta_l$ is sometimes using different fonts. For example, when comparing the abstract to Table 1. Please use consistent font.
6. 79: Probably it is better to say 'lower troposphere' since usually not the whole troposphere is unstable in CAOs.
7. 86-89: Is there a reference for this?
8. 150-153: I'm not sure if I understand how the vertical profile data is actually retrieved since it is written somewhat confusingly. Please try to edit this sentence. Maybe the authors can include these points in Figure 1 as well.
9. 223-224: Can you state any conclusions from this examination? Was any comparison done to test the robustness?
10. Figure 2: In (a) and (d) there is a substantial difference between the BCB height and the actual cloud base. For (a) it looks like that for a higher BCB leg the coupling classification might have been different. Did the authors check how frequent such cases are?
11. 291: I suggest adding the 37.5N line to Figure 3. Also add headings to the sub-figures to indicate which coupling regime they belong to.
12. Table 3 could be removed and the numbers could just be added to Figure 3 to save space.
13. 327-329: Do the authors have an idea of how representative the sampling is compared to climatology? A short sentence about this could be added.
14. 367: Consider using something different than 'difference calculations'. I was confused about this at first. Maybe 'lowest $\Delta$ values'.
15. 381-383 (and in other instances before): I suggest rounding these percentages to full numbers, i.e., 59.70% to 60%; the decimals do not provide much information in this case.
16. 429: The authors should add that this is based on just 4 samples.

**Typographical**

17. 198: Use Greek symbol for 'kappa'

---

## Author Comment (AC1)

We thank the two reviewers for their helpful comments. We have provided our responses to the comments below in blue.

REFEREE 1

Zeider et al. use aircraft measurements from NASA's ACTIVATE field campaign between 2020 and 2022 to investigate aerosol and cloud properties in the marine boundary layer (MBL) for four different coupling regimes. The authors analyze what they refer to as strongly, weakly and two sets of moderately coupled MBLs, instead of only differentiating coupled and decoupled MBLs. These coupling regimes are based on potential temperature and total water mixing ratio differences between cloud base and minimum flight altitude. The authors present findings that are novel for the investigated region and their results highlight the importance of investigating the degree of coupling.

Overall, the paper is well-written, easy to follow, concise and has important findings regarding coupling in the marine boundary layer. Thus, I believe that the manuscript merits publication. Nevertheless, there are some comments mostly regarding the robustness of the results that should be addressed by the authors before publication.

**Response:** Thanks for the positive outlook and comments.

**Major comments**

1. **Impact of measurement uncertainty on results:** Figure S1 shows that many data points are very close to the boundary of their regime. Given measurement uncertainties it is possible that some data points belong to a different regime. I suggest that the authors test the sensitivity of their results by varying the $\Delta\theta_l$ and $\Delta q_t$ values (leading to different regimes for some data) based on measurement uncertainty. Other approaches to account for the measurement uncertainty would be possible as well. I want to add that I found Figure S1 more informative than Figure 4. If the authors want to retain Figure 4, I suggest using white color for pixels with no data points.

**Response:** This is a great comment – thank you. First, we swapped Figure S1 with Figure 4, so now Figure S1 is in the main text and Figure 4 is in the SI document. Second, we performed sensitivity test results by varying $\Delta q_t$ and $\Delta \theta_\ell$ and have included the table shown below in the SI.

While the number of points in each coupling regime did sometimes vary noticeably when changing $\Delta q_t$ and $\Delta \theta_\ell$ values, the means for each property for each coupling regime remain generally the same. Additionally, the trends when comparing means across strong/weak coupling regimes and between the two moderate coupling regimes are almost entirely unchanged (except for a coupling of instances for $\Delta$IntV). The results are consistent even when varying $\Delta q_t$ and $\Delta \theta_\ell$ significantly, such as to 0.5 / 0.5 to match the Jones et al. (2011) and Dong et al. (2015) coupling criteria.

**Table S1.** Sensitivity test results for aerosol and other atmospheric properties investigated across the MinAlt-BCB pairs. $\Delta q_t$ and $\Delta \theta_\ell$ (left/right) were varied (the original criteria are reported in the first column) and the column headers are interpreted as such: for the 0.6/1.0 column, strong coupling is for $\Delta \theta_\ell \leq 1.0$ K and $\Delta q_t \leq 0.6$ g kg$^{-1}$; moderate coupling with high $\Delta \theta_\ell$ is for $\Delta \theta_\ell > 1.0$ K and $\Delta q_t \leq 0.6$ g kg$^{-1}$; moderate coupling with high $\Delta q_t$ is for $\Delta \theta_\ell \leq 1.0$ K and $\Delta q_t > 0.6$ g kg$^{-1}$; and weak coupling is for $\Delta \theta_\ell > 1.0$ K and $\Delta q_t > 0.6$ g kg$^{-1}$. Flight data for MinAlt-BCB pairs were grouped based on the new criteria, and the values reported are the means of the grouped data for each coupling regime. The $\Delta$ calculation refers to the MinAlt value minus the BCB value.

| | 0.8 / 1.0 | 0.6 / 1.0 | 0.7 / 1.0 | 0.9 / 1.0 | 1.0 / 1.0 | 0.8 / 0.8 | 0.8 / 0.9 | 0.8 / 1.1 | 0.8 / 1.2 | 0.5 / 0.5 |
|---|---|---|---|---|---|---|---|---|---|---|
| **# points** | | | | | | | | | | |
| Strong coupling | 293 | 274 | 286 | 302 | 310 | 287 | 289 | 297 | 303 | 210 |
| Moderate coupling, high $\Delta \theta_\ell$ | 56 | 53 | 56 | 57 | 57 | 62 | 60 | 52 | 46 | 92 |
| Moderate coupling, high $\Delta q_t$ | 42 | 61 | 49 | 33 | 25 | 35 | 38 | 42 | 44 | 63 |
| Weak coupling | 20 | 23 | 20 | 19 | 19 | 27 | 24 | 20 | 18 | 46 |
| **$\Delta$scat** | | | | | | | | | | |
| Strong coupling | 2.2 | 1.9 | 2.2 | 2.2 | 2.2 | 2.2 | 2.2 | 2.2 | 2.2 | 2.2 |
| Moderate coupling, high $\Delta \theta_\ell$ | 3.5 | 3.5 | 3.4 | 3.4 | 3.4 | 3.3 | 3.3 | 3.7 | 3.9 | 3.0 |
| Moderate coupling, high $\Delta q_t$ | 2.4 | 2.5 | 2.5 | 2.4 | 2.7 | 2.6 | 2.5 | 2.4 | 2.5 | 2.1 |

| | | | | | | | | | | |
|---|---|---|---|---|---|---|---|---|---|---|
| Weak coupling | 3.5 | 3.3 | 3.5 | 3.5 | 3.5 | 3.0 | 3.2 | 3.5 | 3.4 | 3.1 |
| **ΔIntV** | | | | | | | | | | |
| Strong coupling | 2.5 | 1.5 | 2.5 | 2.5 | 2.5 | 2.5 | 2.5 | 2.5 | 2.5 | 2.5 |
| Moderate coupling, high $\Delta\theta_\ell$ | 2.1 | 2.1 | 2.1 | 2.2 | 2.2 | 2.2 | 2.2 | 2.2 | 2.2 | 2.1 |
| Moderate coupling, high $\Delta q_t$ | 1.9 | 2.5 | 2.1 | 1.9 | 2.3 | 1.8 | 1.8 | 1.9 | 1.9 | 2.5 |
| Weak coupling | 2.8 | 2.8 | 2.8 | 2.6 | 2.6 | 2.6 | 2.7 | 2.8 | 2.9 | 2.8 |
| **ΔN$_{a>3\mu m}$** | | | | | | | | | | |
| Strong coupling | 0.32 | 0.20 | 0.32 | 0.31 | 0.31 | 0.32 | 0.32 | 0.32 | 0.31 | 0.35 |
| Moderate coupling, high $\Delta\theta_\ell$ | 0.33 | 0.35 | 0.33 | 0.32 | 0.32 | 0.32 | 0.32 | 0.34 | 0.38 | 0.29 |
| Moderate coupling, high $\Delta q_t$ | 0.15 | 0.15 | 0.15 | 0.16 | 0.15 | 0.15 | 0.15 | 0.15 | 0.15 | 0.22 |
| Weak coupling | 0.53 | 0.45 | 0.53 | 0.57 | 0.57 | 0.41 | 0.46 | 0.53 | 0.56 | 0.31 |
| **N$_d$** | | | | | | | | | | |
| Strong coupling | 344 | 366 | 346 | 344 | 348 | 345 | 345 | 343 | 343 | 356 |
| Moderate coupling, high $\Delta\theta_\ell$ | 419 | 422 | 421 | 419 | 419 | 411 | 412 | 432 | 441 | 376 |
| Moderate coupling, high $\Delta q_t$ | 329 | 318 | 334 | 343 | 294 | 373 | 362 | 345 | 336 | 371 |
| Weak coupling | 275 | 279 | 275 | 270 | 270 | 263 | 267 | 275 | 286 | 254 |
| **MinAlt $\sigma_w$** | | | | | | | | | | |
| Strong coupling | 0.86 | 1.17 | 0.87 | 0.86 | 0.86 | 0.85 | 0.85 | 0.86 | 0.86 | 0.85 |
| Moderate coupling, high $\Delta\theta_\ell$ | 1.00 | 1.00 | 0.99 | 0.99 | 0.99 | 1.01 | 1.02 | 0.96 | 0.97 | 0.97 |
| Moderate coupling, high $\Delta q_t$ | 0.81 | 0.78 | 0.77 | 0.73 | 0.79 | 0.84 | 0.84 | 0.81 | 0.80 | 0.86 |
| Weak coupling | 0.49 | 0.58 | 0.55 | 0.52 | 0.52 | 0.57 | 0.53 | 0.55 | 0.53 | 0.63 |
| **BCB $\sigma_w$** | | | | | | | | | | |
| Strong coupling | 0.70 | 0.86 | 0.71 | 0.70 | 0.69 | 0.71 | 0.71 | 0.70 | 0.71 | 0.71 |
| Moderate coupling, high $\Delta\theta_\ell$ | 0.64 | 0.68 | 0.64 | 0.65 | 0.65 | 0.64 | 0.64 | 0.64 | 0.61 | 0.67 |
| Moderate coupling, high $\Delta q_t$ | 0.81 | 0.72 | 0.75 | 0.83 | 0.99 | 0.79 | 0.82 | 0.81 | 0.81 | 0.67 |
| Weak coupling | 0.49 | 0.44 | 0.49 | 0.48 | 0.48 | 0.60 | 0.54 | 0.49 | 0.47 | 0.72 |
| **BCB - MinAlt $\sigma_w$** | | | | | | | | | | |
| Strong coupling | -0.15 | -0.31 | -0.15 | -0.16 | -0.16 | -0.15 | -0.14 | -0.16 | -0.16 | -0.15 |
| Moderate coupling, high $\Delta\theta_\ell$ | -0.34 | -0.32 | -0.35 | -0.34 | -0.34 | -0.37 | -0.38 | -0.33 | -0.36 | -0.30 |
| Moderate coupling, high $\Delta q_t$ | 0.01 | -0.06 | -0.02 | 0.10 | 0.20 | -0.04 | -0.03 | 0.01 | 0.01 | -0.19 |

| Weak coupling | -0.05 | -0.14 | -0.05 | -0.05 | -0.05 | 0.03 | 0.01 | -0.05 | -0.06 | 0.09 |
|---|---|---|---|---|---|---|---|---|---|---|

We also added this text to the main article file to address this new addition:

"We also note that sensitivity tests were conducted (Table S1) to see how the assignment of MinAlt-BCB pairs to the four coupling categories changed when accounting for measurement uncertainties, which could push points across the border of their regime in Fig. 4. Results are preserved with only slight changes in assignments after varying the criteria for $\Delta q_t$ and $\Delta \theta_\ell$ by absolute values of 0.02 in both directions; the same applies using the Jones et al. (2011) criteria, which is shown on the far right of Table S1. Subsequent effects on other results presented in the following sections were minimal with the same general conclusions reached."

2. **Missing significance testing in section 3.2:** In section 3.3 the authors test the statistical significance of their results, however, this is missing in section 3.2. For some parameters, standard deviations are relatively large and in general the sample sizes of the non-strongly coupled regimes is quite small. I suggest conducting statistical significance tests for the values in Table 4. At least for the comparisons done in the text some level of statistical significance should be mentioned. Instead of displaying the data in Table 4, the authors could also consider presenting this data similar to the display shown in Figure 5.

**Response:** Thank you for this comment. We have generated another figure for the data from Table 3 similar to Figure 5 (Figure S2) and have added additional text based on these results. Please see the figure and additions below:

[Figure]

**Figure S2**. Notched box plots of various atmospheric properties investigated across MinAlt-BCB pairs. Refer to Table 3 for additional statistics and total number of points. The box plots are colored according to degree of coupling: blue (strong), red (moderate, high $\Delta\theta_\ell$), yellow (moderate, high $\Delta q_t$), and purple (weak). The notches of the boxes assist in the determination of statistical significance between multiple medians (the shading indicates where the notches begin and end). If notches/shading do not overlap, the medians are statistically different from one another (also referred to as statistically significant).

Text added: "The results of the aerosol and atmospheric parameter calculations across the four different coupling regimes are provided in Table 3 (seasonal results in Tables S2-S3), with notched box plots summarizing information from Table 3 in Fig. S2."

"Further, while BCB $\sigma_w$ and BCB – MinAlt $\sigma_w$ had no significant differences across medians, there were some significant differences across coupling regimes for MinAlt $\sigma_w$ (Figure S2). Data in the moderate, high $\Delta\theta_\ell$ coupling regime was significantly different from the other regimes,

and data categorized as moderate coupling with high $\Delta q_t$ were also statistically distinct from the weak coupling regime."

"The results based on medians agree with those of mean values in Table 3, although medians across regimes for each atmospheric property were not statistically different from one another (Fig. S2)."

3. **Moderate coupling cases with very large $\Delta\theta_l$**: Figure S1 and 4 show moderately coupled cases with $\Delta\theta_l$ of 5-10 K. For such a substantial difference of $\theta_l$ can one still speak of a (moderately) coupled boundary layer? I guess what I am wondering is whether the data from these (admittedly) few points is consistent with the rest of the data in this coupling regime?

**Response:** This is a great question. Below are the five points that fall within that 5-10 K $\Delta\theta_\ell$ data range for the moderately coupled, high $\Delta\theta_\ell$ regime. The bolded values are the reported maxima of each atmospheric parameter from the new and revised Table 3. Generally, most of the values are around the median of the datasets. The exception to this rule is $\Delta$scat, as the values below fall toward the max of the dataset for this coupling regime. We feel as though these data points are generally consistent with the coupling regime, which is why they did not get filtered out during the original data scrubbing.

Before analyzing the results reported in the main text, we searched the data for any extreme outliers that were skewing the data and did not seem to align with other values for the various atmospheric parameters. The data used in the paper and also those selected below passed that original test. Therefore, we have not made any changes to the paper based upon this comment.

| $\Delta q_t$ / $\Delta\theta_\ell$ | 0.8 / 1.0 | -2.14 / 10.59 | -1.11 / 9.85 | -0.54 / 9.70 | -0.51 / 5.64 | -1.80 / 5.14 |
|---|---|---|---|---|---|---|
| $\Delta$scat | 3.5 | **14.6** | | 8.4 | 11.6 | 13.3 |
| $\Delta$IntV | 2.1 | 1.3 | | 0.5 | 0.7 | **9.3** |
| $\Delta N_{a>3\mu m}$ | 0.33 | 0.47 | 0.13 | | 0.01 | 1.06 |
| $N_d$ | 419 | 252 | 252 | 330 | 126 | 224 |

| | | | | | | |
|---|---|---|---|---|---|---|
| **MinAlt** $\sigma_w$ | 1.00 | 1.18 | **1.28** | 0.14 | 0.66 | 0.32 |
| **BCB** $\sigma_w$ | 0.64 | 0.31 | 0.35 | 0.32 | 0.88 | 0.41 |
| **BCB - MinAlt** $\sigma_w$ | -0.34 | -0.88 | -0.93 | 0.19 | 0.22 | 0.09 |

**Specific Comments**

4. While most of the paper is well-written, I found the abstract a bit hard to read on the first read. There a very long and wordy sentences. It might make sense to revisit the abstract and edit it for easier readability.

**Response**: We have made the following changes to the abstract for clarity:

"Quantifying the degree of coupling between marine boundary layer clouds and the surface is critical for understanding the evolution of low clouds and explaining the vertical distribution of aerosols and microphysical cloud properties. Previous work has characterized the boundary layer as either coupled or decoupled but this study rather considers four degrees of coupling, ranging from strongly to weakly coupled. We use aircraft data from the NASA Aerosol Cloud meTeorology Interactions oVer western ATlantic Experiment (ACTIVATE) to assess aerosol and cloud characteristics for the following four regimes, quantified using differences in liquid water potential temperature ($\theta_\ell$) and total water mixing ratio ($q_t$) between flight data near-surface level (~150 m) and directly below cloud bases: strong coupling ($\Delta\theta_\ell \leq 1.0$ K, $\Delta q_t \leq 0.8$ g kg$^{-1}$), moderate coupling with high $\Delta\theta_\ell$ ($\Delta\theta_\ell > 1.0$ K, $\Delta q_t \leq 0.8$ g kg$^{-1}$), moderate coupling with high $\Delta q_t$ ($\Delta\theta_\ell \leq 1.0$ K, $\Delta q_t > 0.8$ g kg$^{-1}$), weak coupling ($\Delta\theta_\ell > 1.0$ K, $\Delta q_t > 0.8$ g kg$^{-1}$). Results show that (i) turbulence is greater in the strong coupling regime compared to the weak coupling regime, with the former corresponding to more vertical homogeneity in 550 nm aerosol scattering, integrated aerosol volume concentration, and giant aerosol number concentration ($D_p > 3$ µm) coincident with increased MBL mixing; (ii) cloud drop number concentration is greater during periods of strong coupling due to the greater upward vertical velocity and subsequent activation of particles; (iii) sea-salt tracer species ($Na^+$, $Cl^-$, $Mg^{2+}$, $K^+$) are present in greater concentrations in the strong coupling regime compared to weak coupling, while tracers of continental pollution ($Ca^{2+}$, nss-$SO_4^{2-}$, $NO_3^-$, oxalate, and $NH_4^+$) are higher in mass fraction for the weak coupling regime. Additionally, pH and $Cl^-$:$Na^+$ (a marker for chloride depletion) are consistently lower in the weak coupling regime. There were also differences between the two moderate regimes: the moderate,

high $\Delta q_t$ regime had greater turbulent mixing and sea salt concentrations in cloud water, along with smaller differences in integrated volume and giant aerosol number concentration across the two vertical levels compared. This work shows value in defining multiple coupling regimes (rather than the traditional coupled versus decoupled) and demonstrates differences in aerosol and cloud behavior in the MBL for the various regimes."

5. $\Delta\theta_l$ is sometimes using different fonts. For example, when comparing the abstract to Table 1. Please use consistent font.

**Response:** Changes made.

6. 79: Probably it is better to say 'lower troposphere' since usually not the whole troposphere is unstable in CAOs.

**Response:** Change made.

7. 86-89: Is there a reference for this?

**Response:** Yes, it is the same reference as the previous sentence but we now explicitly added the Stevens et al. (1998) reference to the end of this sentence in question:

"The latter case can be viewed as weakly coupled due to episodic updraft-driven convection that is less efficient at mixing the MBL than is the case in well-mixed MBLs in which downdrafts associated with cloud-top radiative cooling couple the cloud and sub-cloud layers (Stevens et al., 1998)."

8. 150-153: I'm not sure if I understand how the vertical profile data is actually retrieved since it is written somewhat confusingly. Please try to edit this sentence. Maybe the authors can include these points in Figure 1 as well.

**Response:** We have updated Figure 1 to better indicate where the data for the vertical profile are taken and have updated the figure caption as well as added additional text:

[Figure]

**Figure 1**: Cloudy ensemble flight strategy of the HU-25 Falcon during the ACTIVATE flights, where the grey box represents a typical cloud layer with upper and lower boundaries representing cloud top and base, respectively. The order of legs was the nominal plan that was flown, but sometimes the legs were flown in a different order based on flight restrictions and cloud conditions. The red star indicates where the BCB level would be marked for this particular flight pattern, which is during the slant descent from ACB to MinAlt and uses the mean altitude of the preceding BCB leg immediately before the ACB leg; that level would then be compared to the adjacent MinAlt level that begins at the end of the slant descent. Otherwise, MinAlt-BCB pairs that are used include when a MinAlt level leg was immediately preceded or succeeded by a BCB level leg. The green line illustrates the data that would be used to investigate the vertical structure of the layer, starting with the last timestamp from the pseudo-BCB leg and ending with the first timestamp in the MinAlt leg.

Revised text: "The vertical structure of the layer between MinAlt and BCB was examined using data between the last time stamp in the MinAlt/BCB leg (i.e., whichever was first in the MinAlt-BCB pair) and first-time stamp in the subsequent BCB/MinAlt leg (i.e., whichever was second in the MinAlt-BCB pair). This is indicated by a green line on Fig. 1, which begins with the last time

stamp from the pseudo-BCB leg (indicated by the red star) and ends with the first time stamp in the MinAlt leg."

9. 223-224: Can you state any conclusions from this examination? Was any comparison done to test the robustness?

**Response:** We visually examined the profiles just to confirm there was nothing very unusual occurring in the vertical profiles. We reviewed past references showing such profiles and we felt the level of detail we provided was sufficient to be on par with those other studies. As a result, we did not feel additional text or analysis was needed for this comment.

10. Figure 2: In (a) and (d) there is a substantial difference between the BCB height and the actual cloud base. For (a) it looks like that for a higher BCB leg the coupling classification might have been different. Did the authors check how frequent such cases are?

**Response:** These cases were not too common. Even in these more extreme cases in (a) and (d) with a relatively larger gap between the BCB leg and cloud base, the assignment to a particular coupling category would not have changed. We did rigorously look into these factors to ensure the assignments were done well. The nature of the flights was such that it was not easy always to position the BCB perfectly right below the cloud base height and sometimes there was a bit of a gap. We did add this text to the paper:

"We note that while Fig. 2a/d show BCB legs being relatively farther below the actual cloud base height (Fig. 2a = 223 m; Fig. 2d = 370 m) than the other two examples (Fig. 2b = 101 m; Fig. 2c = 30 m), the former two were anomalous cases and usually the BCB legs were closer to cloud base. As noted by Dadashazar et al. (2022b), the Falcon aimed to conduct BCB and ACB legs about ~100 m below and above the estimated cloud base height, respectively. Median/mean distances from BCB to cloud bases were as follows for all samples in the four coupling categories: strong = 73/87 m; moderate, high $\Delta\theta_\ell$ = 101/119 m; moderate, high $\Delta q_t$ = 69/71 m; weak = 104/142 m."

11. 291: I suggest adding the 37.5N line to Figure 3. Also add headings to the sub-figures to indicate which coupling regime they belong to.

**Response:** This is a useful suggestion. We have added an orange dashed line on Figure 3 to indicate 37.5°N and updated the figure caption. Additionally, we have added sub-figure headings. Please see the following figure:

[Figure]

**Figure 3:** Locations of the BCB segments of the MinAlt-BCB pairs (blue circles), broken up into the four different degrees of coupling. The MinAlt legs were close in time to the BCB legs, so only one spatial map is needed to show the approximate data point location for each pair. The locations of the cloud water samples (white triangles) are overlaid on the BCB segment locations. The black star indicates the location of NASA Langley Research Center, the red diamond indicates Bermuda, and the orange dashed line indicates 37.5°N, which is referenced in the discussion about this figure. The total number of MinAlt-BCB pairs for each category are also included for each coupling regime. The winter months include January, February, March, and April, while the summer months include May, June, August, and September.

12. Table 3 could be removed and the numbers could just be added to Figure 3 to save space.

**Response:** This is a great comment. We have added the numbers from Table 3 into Figure 3. Please refer to the previous comment for the added figure and updated caption.

13. 327-329: Do the authors have an idea of how representative the sampling is compared to climatology? A short sentence about this could be added.

**Response:** This is an excellent question but a hard one to answer with enough confidence in our view to warrant a sentence about it. All field campaigns have limitations but we firmly believe ACTIVATE did one of the best jobs to date to fly a region in a representative way to get close to climatology. We have revised the text in question a bit to keep it concise and to the point; it's hard to quantify an answer to the reviewer's question:

"While ACTIVATE flights were designed for achieving a statistically rich dataset portraying the region in an unbiased way, the frequency of occurrence of the four coupling regimes in this study can possibly still be affected by how flights were designed to fly towards areas with relatively higher cloud fraction without complicating scenes such as with multiple cloud layers."

14. 367: Consider using something different than 'difference calculations'. I was confused about this at first. Maybe 'lowest Δ values'.

**Response:** Change made.

15. 381-383 (and in other instances before): I suggest rounding these percentages to full numbers, i.e., 59.70% to 60%; the decimals do not provide much information in this case.

**Response:** Changes made.

16. 429: The authors should add that this is based on just 4 samples.

**Response:** Addressed with this addition:

"This is consistent with the aerosol results in Sect. 3.2 that suggested this latter category can have appreciable influence from the surface, although it is also important to note that this is based on 4 cloud water samples."

**Typographical**

17. 198: Use Greek symbol for 'kappa'

**Response:** Change made.

REFEREE 2

This is an interesting study of the variation of cloud and aerosol properties over the northwest Atlantic region using observations during the ACTIVATE campaign. By carefully selecting aircraft measurements in specific flight patterns, this manuscript reports different aerosol and cloud properties in environments with different boundary strengths. The manuscript is well-written, and I do not have major concerns about the reported results. I have the following comments for the authors to consider.

**Response:** We thank the reviewer for the positive feedback.

**Major**

While it is clear how the cloud and aerosol properties differ under different boundary layer conditions, the motivation of this work is not well conceived. I agree that having multiple categories to describe boundary layer coupling strength is important, but how do the proposed categorization and the reported results help the modeling community? Instead of more categories (mainly the moderate coupled categories) and a name is 'weakly coupled' rather than 'decoupled' boundary layer, what are the new findings from the boundary layer coupling strength analysis in this work? Or is it to confirm some of the previous findings in other regions that are also true in the northwest Atlantic?

**Response:** This is a fair concern and one we thought about hard before we even began the study since we did not want to aimlessly work on something that wouldn't matter for the community.

In the last paragraph of the introduction we tried to motivate why our choice to introduce 4 coupling categories would be useful and why we did the full analyses we did. We don't think the usefulness of this paper is most geared towards modeling but instead it is important for basic data analyses studies and can help with future flight designs. Recall that we originally had this in the text:

"The analyses presented here are important for reasons such as knowing how well the aerosol near the surface level represent the aerosol just below cloud bases, with implications for the aerosol that largely govern the drop concentration budget."

Also, we did have this text before that hopefully helps (which we slightly revised):

"Differences identified in aerosol and cloud characteristics between these four coupling regimes are important to inform both future flight designs and data analysis research to account for thermodynamic profiles when examining aspects of aerosol and cloud microphysics when using either satellite, reanalysis, airborne, or ground-based datasets."

Another achievement in this study is that we confirmed the hypotheses stated in that last paragraph of the Introduction too.

**Minor**

L106: what do you mean by 'more similar values for aerosol properties in the sub-cloud layer'? Are you referring to, for example, the aerosol numbers and compositions are similar among coupled cases?

**Response:** In this case, we are referring to aerosol number and composition, but also forthcoming metrics such as aerosol scattering and integrated volume. We have revised this sentence for clarity:

Revised text: "(i) more vertical homogeneity in aerosol microphysical properties such as number concentration between the sub-cloud layer and closer to the ocean's surface…"

L106-108: Please clarify if the statements in the list are from previous studies and preferably include references for each statement. I am curious about what the third statement means. Higher cloud droplet number concentration compared to where or what? Compare to less coupled cases?

**Response:** Fixed:

"Although there are scarce previous reports of such findings (e.g., Dong et al., 2015; Wang et al., 2016), the central hypotheses are based on confirming what has been shown or implied in other regions, in that more strongly coupled cases will have (i) more vertical homogeneity in aerosol properties between the sub-cloud layer and closer to the ocean's surface, (ii) cloud composition reflecting significantly more sea salt influence (Wang et al., 2016), and (iii) higher cloud drop number concentration, compared to weakly coupled cases (Dong et al., 2015)."

L150-153: It took me a while to figure out how the data was analyzed. Can you mark the portion of the legs where the data is used in Figure 1?

**Response:** We have updated Figure 1 as well as added additional text to better illustrate where the data for the vertical profile were taken. Please refer to Referee 1 Minor Comment 8 for the updated figure and text revisions made.

L155: do you restrict how far away (or how many seconds apart) the two adjacent MinAlt-BCB legs are?

**Response:** Our flight strategy was essentially the same for flight times when we used these so-called adjacent MinAlt-BCB legs. Basically one leg is done, and there is a slant descent/ascent to the next leg. The time lag is ~2 minutes. As this was routine, no extra restrictions were needed. We don't feel any text is needed to address this comment.

L156: can you clarify, in the case of slants, if the data are taken from around the BCB leg or the red star in Figure 1? I assume it is the former, but I'm not sure.

**Response:** We have made the following changes to the text:

"This study compares various measurement data (Sect. 2.5) between MinAlt and BCB legs using the last/first 5 seconds of data during adjacent MinAlt-BCB legs, and in the case of slants, we use the 2 points before and after the actual BCB point (the red star in Fig. 1) for a total of 5 points (average altitude range ~ 20 m) that represent the BCB level with the condition that all data were out of cloud."

L219: How do you determine the threshold for ? Is it based on visual observations of the coupling state and statistics of all the profiles collected during ACTIVATE? Does the distance between BCB and the actual cloud base height affect the threshold selection?

**Response:** This is a difficult task to determine a threshold as there is no perfect way to do it. We selected thresholds in such a way to be informed by the literature and to not deviate too much from values in Table 1. We examined our vertical profiles and statistics and felt it was prudent to use the four sets of criteria presented in the paper. Thus, we are informed by literature and the region's vertical profile statistics. As noted in response to a previous reviewer comment, Figures 2a/d are more extreme cases of the BCB height being relatively far below the cloud base height, but even in those cases the categorization remains unaffected. There is some slight differences in the final calculated $\Delta q_t$ and $\Delta \theta_\ell$ based on how far the BCB height is from cloud base height, but we feel confident that our general conclusions and results are unaffected by this issue. Thus, we do not feel we need to add any text to address this comment in the paper.

Figure 2: I understand this is a typical case, but do most weakly coupled cases have BCB legs around the middle of the cloud-base height?

**Response:** We had a little trouble understanding this comment since the weakly coupled cases showed the BCB legs being below the cloud base height. Perhaps there was a misunderstanding on how to interpret Figure 2. Or maybe they meant that the BCB leg was between the halfway point of the cloud base and bottom of the plot? In that case, we felt it was helpful to add this text:

"We note that while Fig. 2a/d show BCB legs being relatively farther below the actual cloud base height (Fig. 2a = 223 m; Fig. 2d = 370 m) than the other two examples (Fig. 2b = 101 m;

Fig. 2c = 30 m), these were anomalous cases and usually the BCB legs were closer to cloud base. As noted by Dadashazar et al. (2022b), the Falcon aimed to conduct BCB and ACB legs about ~100 m below and above the estimated cloud base height, respectively. Median/mean distances from BCB to cloud bases were as follows for all samples in the four coupling categories: strong = 73/87 m; moderate, high $\Delta\theta_\ell$ = 101/119 m; moderate, high $\Delta q_t$ = 69/71 m; weak = 104/142 m."

L338-340: this is related to the question on the height of CBC leg altitude. Suppose it is the case that the CBC usually sits around half of the cloud-base height. In that case, you are calculating vertical velocity variance at different levels of the boundary layer for the weakly coupled cases than for the other three categories.

**Response:** Thanks for the good comment. The weakly coupled case in Figure 2d is a bit unusual in that the BCB level leg was quite a bit below cloud base height but that was not usually the case with most BCB legs being closer to the cloud base height. See response to previous comment. We don't feel text is needed to address this comment as not all cases were exactly like those in Figure 2.

L351-352: here, the giant particles refer to aerosol particles, right?

**Response:** Correct. We don't feel any changes are needed for this comment.

---

## Author Comment (AC2)

We thank the two reviewers for their helpful comments. We have provided our responses to the comments below in blue.

REFEREE 1
Second review of Sensitivity of aerosol and cloud properties to coupling strength of marine boundary layer clouds over the northwest Atlantic by Zeider et al. (2024)

The authors overall did a good job addressing the other reviewer's and my comments. The manuscript has improved, and I can recommend it for publication, but suggest that the authors consider the additional comments I have made below. Please note that line numbers are based on the tracked changes manuscript.

1. Regarding my first major comment, it is not entirely clear whether the chosen variation of the thresholds makes sense in terms of measurement uncertainty. The authors might want to consider explicitly mentioning values for the measurement uncertainty. Furthermore, I think sensitivity tests 0.6/0.8 and 1.0/1.2 could be included varying both parameters at the same time. The tests with small variations (0.7/1.0, 0.9/1.0, 0.8/0.9, 0.8/1.1) could be omitted since the tests with larger variations (0.6/1.0, etc.) are also consistent with the main results. At the moment, the analysis the authors provide appears to me to mostly test the robustness of the (somewhat arbitrarily) chosen thresholds but not explicitly account for a known uncertainty of the measurements.

**Response:** We have explicitly listed measurement uncertainties in Table 2 and made mention of them in the text when discussing the sensitivity tests of Table S1:

"We also note that sensitivity tests were conducted (Table S1) to see how the assignment of MinAlt-BCB pairs to the four coupling categories changed when accounting for measurement uncertainties (shown in Table 2), which could push points across the border of their regime in Fig. 4. Varying $\Delta q_t$ and $\Delta \theta_\ell$ by absolute values of 0.2 in both directions was investigated to test for sensitivity to measurement uncertainty in this study."

Also, we test the two situations the reviewer mentioned and added them to Table S1 without much change to the story. We didn't feel we needed to omit the old combinations we tested as it doesn't hurt to include them along with the new ones.

| | 0.8/1.0 | 0.6/1.0 | 0.7/1.0 | 0.9/1.0 | 1.0/1.0 | 1.0/1.2 | 0.8/0.8 | 0.8/0.9 | 0.8/1.1 | 0.8/1.2 | 0.6/0.8 | 0.5/0.5 |
|---|---|---|---|---|---|---|---|---|---|---|---|---|
| **# points** | | | | | | | | | | | | |
| Strong coupling | 293 | 274 | 286 | 302 | 310 | 320 | 287 | 289 | 297 | 303 | 268 | 210 |

| | | | | | | | | | | | | |
|---|---|---|---|---|---|---|---|---|---|---|---|---|
| Moderate coupling, high $\Delta\theta_\ell$ | 56 | 53 | 56 | 57 | 57 | 47 | 62 | 60 | 52 | 46 | 59 | 92 |
| Moderate coupling, high $\Delta q_t$ | 42 | 61 | 49 | 33 | 25 | 27 | 35 | 38 | 42 | 44 | 54 | 63 |
| Weak coupling | 20 | 23 | 20 | 19 | 19 | 17 | 27 | 24 | 20 | 18 | 30 | 46 |
| **Δscat** | | | | | | | | | | | | |
| Strong coupling | 2.2 | 1.9 | 2.2 | 2.2 | 2.2 | 2.2 | 2.2 | 2.2 | 2.2 | 2.2 | 2.2 | 2.2 |
| Moderate coupling, high $\Delta\theta_\ell$ | 3.5 | 3.5 | 3.4 | 3.4 | 3.4 | 3.9 | 3.3 | 3.3 | 3.7 | 3.9 | 3.4 | 3.0 |
| Moderate coupling, high $\Delta q_t$ | 2.4 | 2.5 | 2.5 | 2.4 | 2.7 | 2.8 | 2.6 | 2.5 | 2.4 | 2.5 | 2.6 | 2.1 |
| Weak coupling | 3.5 | 3.3 | 3.5 | 3.5 | 3.5 | 3.4 | 3.0 | 3.2 | 3.5 | 3.4 | 2.9 | 3.1 |
| **ΔIntV** | | | | | | | | | | | | |
| Strong coupling | 2.5 | 1.5 | 2.5 | 2.5 | 2.5 | 2.4 | 2.5 | 2.5 | 2.5 | 2.5 | 2.4 | 2.5 |
| Moderate coupling, high $\Delta\theta_\ell$ | 2.1 | 2.1 | 2.1 | 2.2 | 2.2 | 2.3 | 2.2 | 2.2 | 2.2 | 2.2 | 2.2 | 2.1 |
| Moderate coupling, high $\Delta q_t$ | 1.9 | 2.5 | 2.1 | 1.9 | 2.3 | 2.3 | 1.8 | 1.8 | 1.9 | 1.9 | 2.6 | 2.5 |
| Weak coupling | 2.8 | 2.8 | 2.8 | 2.6 | 2.6 | 2.6 | 2.6 | 2.7 | 2.8 | 2.9 | 2.6 | 2.8 |
| **$\Delta N_{a>3\mu m}$** | | | | | | | | | | | | |
| Strong coupling | 0.32 | 0.20 | 0.32 | 0.31 | 0.31 | 0.30 | 0.32 | 0.32 | 0.32 | 0.31 | 0.33 | 0.35 |
| Moderate coupling, high $\Delta\theta_\ell$ | 0.33 | 0.35 | 0.33 | 0.32 | 0.32 | 0.37 | 0.32 | 0.32 | 0.34 | 0.38 | 0.34 | 0.29 |
| Moderate coupling, high $\Delta q_t$ | 0.15 | 0.15 | 0.15 | 0.16 | 0.15 | 0.15 | 0.15 | 0.15 | 0.15 | 0.15 | 0.16 | 0.22 |
| Weak coupling | 0.53 | 0.45 | 0.53 | 0.57 | 0.57 | 0.60 | 0.41 | 0.46 | 0.53 | 0.56 | 0.36 | 0.31 |
| **$N_d$** | | | | | | | | | | | | |
| Strong coupling | 344 | 366 | 346 | 344 | 348 | 348 | 345 | 345 | 343 | 343 | 351 | 356 |
| Moderate coupling, high $\Delta\theta_\ell$ | 419 | 422 | 421 | 419 | 419 | 439 | 411 | 412 | 432 | 441 | 412 | 376 |
| Moderate coupling, high $\Delta q_t$ | 329 | 318 | 334 | 343 | 294 | 285 | 373 | 362 | 345 | 336 | 767 | 371 |
| Weak coupling | 275 | 279 | 275 | 270 | 270 | 280 | 263 | 267 | 275 | 286 | 267 | 254 |
| **MinAlt $\sigma_w$** | | | | | | | | | | | | |
| Strong coupling | 0.86 | 1.17 | 0.87 | 0.86 | 0.86 | 0.86 | 0.85 | 0.85 | 0.86 | 0.86 | 0.86 | 0.85 |
| Moderate coupling, high $\Delta\theta_\ell$ | 1.00 | 1.00 | 0.99 | 0.99 | 0.99 | 0.97 | 1.01 | 1.02 | 0.96 | 0.97 | 1.02 | 0.97 |
| Moderate coupling, high $\Delta q_t$ | 0.81 | 0.78 | 0.77 | 0.73 | 0.79 | 0.79 | 0.84 | 0.84 | 0.81 | 0.80 | 0.80 | 0.86 |
| Weak coupling | 0.49 | 0.58 | 0.55 | 0.52 | 0.52 | 0.50 | 0.57 | 0.53 | 0.55 | 0.53 | 0.60 | 0.63 |
| **BCB $\sigma_w$** | | | | | | | | | | | | |

| | | | | | | | | | | | |
|---|---|---|---|---|---|---|---|---|---|---|---|
| Strong coupling | 0.70 | 0.86 | 0.71 | 0.70 | 0.69 | 0.70 | 0.71 | 0.71 | 0.70 | 0.71 | 0.72 | 0.71 |
| Moderate coupling, high $\Delta\theta_\ell$ | 0.64 | 0.68 | 0.64 | 0.65 | 0.65 | 0.62 | 0.64 | 0.64 | 0.64 | 0.61 | 0.67 | 0.67 |
| Moderate coupling, high $\Delta q_t$ | 0.81 | 0.72 | 0.75 | 0.83 | 0.99 | 0.97 | 0.79 | 0.82 | 0.81 | 0.81 | 0.70 | 0.67 |
| Weak coupling | 0.49 | 0.44 | 0.49 | 0.48 | 0.48 | 0.45 | 0.60 | 0.54 | 0.49 | 0.47 | 0.55 | 0.72 |
| **BCB - MinAlt $\sigma_w$** | | | | | | | | | | | | |
| Strong coupling | -0.15 | -0.31 | -0.15 | -0.16 | -0.16 | -0.17 | -0.15 | -0.14 | -0.16 | -0.16 | -0.14 | -0.15 |
| Moderate coupling, high $\Delta\theta_\ell$ | -0.34 | -0.32 | -0.35 | -0.34 | -0.34 | -0.36 | -0.37 | -0.38 | -0.33 | -0.36 | -0.35 | -0.30 |
| Moderate coupling, high $\Delta q_t$ | 0.01 | -0.06 | -0.02 | 0.10 | 0.20 | 0.19 | -0.04 | -0.03 | 0.01 | 0.01 | -0.10 | -0.19 |
| Weak coupling | -0.05 | -0.14 | -0.05 | -0.05 | -0.05 | -0.05 | 0.03 | 0.01 | -0.05 | -0.06 | -0.05 | 0.09 |

2. The authors could include indications of the coupling regime in Figures 5 and S2 directly in the respective figure not just in the caption (i.e. a legend for the colors and/or x-axis labels).

**Response:** Changes made as shown here (Figures 5 then S2, respectively):

[Figure]

[Figure]

3. I feel some of the figure captions are overly long. The authors could consider limiting the captions to be purely descriptive of the figure, any analysis or description of methodology should be in the main text.

**Response:** We have trimmed several figure captions, notably for Figures 1, 3, 5. Please see the revised captions below; the trimmed text was placed back into the main text as suggested by the reviewer.

"**Figure 1:** Cloudy ensemble flight strategy of the HU-25 Falcon during the ACTIVATE flights, where the grey box represents a typical cloud layer. The red star indicates where the BCB level would be marked and the data that would be utilized for this particular flight pattern. Otherwise, MinAlt-BCB pairs that are used include when a MinAlt level leg was immediately preceded or succeeded by a BCB level leg. The green line illustrates the data that would be used to investigate the vertical structure of the layer, starting with the last timestamp from the pseudo-BCB leg and ending with the first timestamp in the MinAlt leg."

"**Figure 3**: Locations of the BCB segments of the MinAlt-BCB pairs (blue circles), broken up into the four different degrees of coupling. The locations of the cloud water samples (white triangles) are overlaid on the BCB segment locations. The black star indicates the location of NASA Langley Research Center, the red diamond indicates Bermuda, and the orange dashed line indicates 37.5°N, which is referenced in the discussion about this figure. The total number of MinAlt-BCB pairs for each category are also included for each coupling regime."

"**Figure 5**: Notched box plots of species concentrations (µg m-3), Cl-:Na+ mass ratio, and pH from cloud water samples collected during periods coinciding with MinAlt-BCB pairs."

4. A few comments regarding your response to my comment 10. If these cases are uncommon or in other words less representative, why were they specifically chosen to be shown in Figure 2 instead of cases that are more representative? A follow-up question I have is, even if coupling regimes were not impacted, is it not possible that other measurements (Δscat, etc.) could have changed significantly if taken closer to cloud base? I guess this question cannot be answered since there is no level leg closer to cloud base for these cases. Given the uncertainty arising from this and these cases being uncommon, would it not make sense to remove them from the analysis since it would have little impact on the results (few cases), making the results more robust (removing cases with larger uncertainty)?

**Response:** The choice of the original profiles was based on having chosen these very early on and we just stuck to them. We certainly are happy to change them out and agree now that this will assist for readers. We kept all the cases as these are the ones that reflect the ACTIVATE dataset, which we will be used by others in the future in the context of comparing BCB legs to in-cloud legs. Rather than cherry-pick the "good ones" where BCB is much closer to ACB, we felt better to use all of our cases even if BCB was farther below ACB than is typically desirable. These larger gaps for some of the cases reflect the challenge of airborne ACI science in the northwest Atlantic which is challenging, and even probably more challenging than typical stratocumulus cloud decks off the eastern coasts of subtropical continental areas. So we keep the original cases but still revise Figure 2 to address the 2nd reviewers comment; we feel this choice of modifications is the best compromise to improve the study.

5. The lack of statistical significance for most variables in Figure S2 might concern some readers. The authors might want to consider including some further discussion on this, e.g., in the context of the sample sizes.

**Response:** The following text was added:

"Although there is a lack of statistically significant differences between the four coupling regimes for the investigated atmospheric properties, it is important to note that the sample sizes for each regime vary greatly. Therefore, there is more variability within the weak coupling regime with only 20 data points compared to the strong coupling regime with over 200 data points. As this study utilized all of the data at its disposal and there were more strong coupling cases than any other coupling regime, the lack of statistical significance across coupling regimes did not impact the general conclusions of the study."

Typographical

6. 444: 0.2 instead of 0.02

**Response:** Thanks for catching this. Change made.

REFEREE 2
Both reviewers note the large gaps between BCB and cloud base height in Figures 2a and 2d.
The authors claim the two cases are not common, so why not change the cases to the ones that
are more representative, especially considering Figure 2 to be illustrative of the four categories?

**Response:** Change made and here is the updated version of Figure 2: